**RESEARCH**     **Open Access**

# A genome assembly and the somatic genetic and epigenetic mutation rate in a wild long-lived perennial *Populus trichocarpa*

Brigitte T. Hofmeister[1], Johanna Denkena[2], Maria Colomé-Tatché[2,3,4], Yadollah Shahryary[5], Rashmi Hazarika[5,6], Jane Grimwood[7,8], Sujan Mamidi[7], Jerry Jenkins[7], Paul P. Grabowski[7], Avinash Sreedasyam[7], Shengqiang Shu[8], Kerrie Barry[8], Kathleen Lail[8], Catherine Adam[8], Anna Lipzen[8], Rotem Sorek[9], Dave Kudrna[10], Jayson Talag[10], Rod Wing[10], David W. Hall[11], Daniel Jacobsen[12], Gerald A. Tuskan[12], Jeremy Schmutz[7,8], Frank Johannes[5,6*] and Robert J. Schmitz[6,11*]

\* Correspondence: frank@
johanneslab.org; schmitz@uga.edu
[5]Department of Plant Sciences,
Technical University of Munich,
Liesel-Beckmann-Str. 2, Freising,
Germany
[6]Institute for Advanced Study (IAS),
Technical University of Munich,
Lichtenbergstr. 2a, Garching,
Germany
Full list of author information is
available at the end of the article

## Abstract

**Background:** Plants can transmit somatic mutations and epimutations to offspring, which in turn can affect fitness. Knowledge of the rate at which these variations arise is necessary to understand how plant development contributes to local adaption in an ecoevolutionary context, particularly in long-lived perennials.

**Results:** Here, we generate a new high-quality reference genome from the oldest branch of a wild *Populus trichocarpa* tree with two dominant stems which have been evolving independently for 330 years. By sampling multiple, age-estimated branches of this tree, we use a multi-omics approach to quantify age-related somatic changes at the genetic, epigenetic, and transcriptional level. We show that the per-year somatic mutation and epimutation rates are lower than in annuals and that transcriptional variation is mainly independent of age divergence and cytosine methylation. Furthermore, a detailed analysis of the somatic epimutation spectrum indicates that transgenerationally heritable epimutations originate mainly from DNA methylation maintenance errors during mitotic rather than during meiotic cell divisions.

**Conclusion:** Taken together, our study provides unprecedented insights into the origin of nucleotide and functional variation in a long-lived perennial plant.

**Keywords:** Mutation rate, Epimutation rate, Epigenetics, Poplar, DNA methylation

## Background

The significance of somatic mutations, i.e., variations in DNA sequence that occur after fertilization, in long-lived plant and animal species, has been a point of debate and investigation for the past 30 years [1–4]. It has been hypothesized that the evolutionary consequences of such mutations are likely even more profound in woody perennial plants, where undifferentiated meristematic cells produce all above-ground and below-ground structures. As meristems undergo constant cell division throughout the lifetime of a plant, somatic mutations arising in meristems may result in genetic differences being passed onto progeny cells [5–8]. The accumulation of somatic mutations can thus lead to genetic and occasionally also phenotypic divergence among vegetative lineages within the same individual. In trees, for instance, different branches have been shown to differ in their responses to pest and pathogen attack, alternate reactions to drought and/or nutrient availability, or dissimilar demands for photosynthate material, even within the same individual [9]. Beyond the impact of point mutations and small insertions/deletions on gene function, alterations in chromatin structure and DNA methylation might also impact gene expression variation.

Phenotypic variation has been attributed to somatic mutations in several perennial plants, including the derivation of Nectarines in peach [10] and the origin of modern grape cultivars (*Vitis vinifera* L.) [11]. In *Populus tremuloides*, somatic mutations have been hypothesized as the cause for variation in DNA markers among individual ramets of a single genotype [12]. Initial attempts to demonstrate within-tree mosaicism using genetic markers [13] showed at low-resolution that the degree of intra-tree variability was positively correlated with the physical distance between sampled branches. More recently, work in oak (*Quercus rubur*) has documented variation in DNA sequence among an independent sampling of alternate branches from a single genotype [14, 15]. They estimated a fixed mutation rate of $4.2–5.2 \times 10^{-8}$ substitutions per locus per generation, which is only within one order of magnitude of the rate observed in the herbaceous annual plant *Arabidopsis thaliana* [16, 17]. These results are consistent with an emerging hypothesis that the per-unit-time mutation rate of perennials is much lower than in annuals to delay mutational meltdown [18, 19], and this lower rate is accomplished by limiting the number of cell divisions between the meristem and the new branch [20]. Additional recent studies have also revealed similar rates of spontaneous mutations in a range of species including perennials [19]. Regardless of the rate of mutation, the frequency of deleterious mutations in woody plants is high, which is hypothesized to reduce survival of progeny resulting from inbreeding and favor outcrossing as is observed in many forest trees [21, 22].

Similar to genetic mutations, phenotypic variation can be caused by epigenetic variation such as stable changes in cytosine methylation or epimutations [23]. Cytosine methylation is a covalent base modification that is inherited through both mitotic and meiotic cell divisions in plants [24]. It occurs in three sequence contexts, CG, CHG, and CHH (H = A, T, or C), and the pattern and distribution of methylation at these different contexts is predictive of its function in genome regulation [25]. Spontaneous changes in methylation independent of genetic changes can lead to phenotypic changes [26]. Well-characterized examples in plants include the peloric phenotype in toadflax (*Linaria vulgaris*), the colorless non-ripening phenotype in tomato (*Solanum lycopersicum*), and the mantled phenotype in oil palm (*Elaeis guineensis*) [27–29].

Once established, epimutations can stably persist or be inherited across generations. For example, the reversion rate from the colorless non-ripening epimutant allele to wild type is about 1 in 1000 per generation in tomato [28]. Studies in *A. thaliana* mutation accumulation lines have documented that the vast majority (91–99.998%) of methylated regions in the genome are stably inherited across generations; only a small subset of the methylome shows variation among mutation accumulation lines [30–33]. Estimates in *A. thaliana* indicate that the spontaneous methylation gain and loss rates at CG sites are $2.56 \times 10^{-4}$ and $6.30 \times 10^{-4}$ per generation per haploid methylome, respectively [34]. Despite the wealth of knowledge about transgenerational methylation inheritance, very little is known about somatic epimutations, especially in long-lived perennial species. Previous studies have been limited by resolution and time. Heer et al. observed no global methylation changes and no consistent variation in gene body methylation associated with growth conditions of Norway spruce [35]. Several studies have linked stress conditions to differential methylation in perennials but did not look at the stability of methylation after removing the stressor [36, 37]. One exception, Le Gac et al., identified environment-related differentially methylated regions in poplar, but only examined stability across 6 months [38].

Detailed insights into the rate and spectrum of somatic mutations and epimutations are necessary to understand how somatic development of long-lived perennials contribute to population-level variation in an ecoevolutionary context. Here we generated a new high-quality reference genome from the oldest branch of a wild *Populus trichocarpa* tree with two dominant stems which have been evolving independently for approximately 330 years. By sampling multiple, age-estimated branches of this tree, we used a multi-omics approach to quantify age-related somatic changes at the genetic, epigenetic, and transcriptional level. Our study provides the first quantitative insights into how nucleotide and functional variation arise during the lifetime of a long-lived perennial plant.

## Results

### Experimental design for the discovery of somatic genetic and epigenetic variants

A stand of trees was identified near Mount Hood, Oregon, and vegetative samples were collected from over 15 trees as part of an independent study. Of these trees, five were chosen for subsequent analysis and five branches of each tree were identified (Additional file 1: Fig. S1). For each branch, the stem age was determined by coring the main stem at breast height and where the branch meets the stem and the branch age was determined by coring the base of the branch (Fig. 1 and Additional file 1: Fig. S2). Although 25 branches in total were initially sampled, six were excluded from analysis because they were epicormic and age estimates could not be determined. Two other branches had incomplete cores, but ages could be estimated based on radial diameter.

From this, we were specifically interested in tree 13 and tree 14 (Fig. 1). Originally identified as two separate genotypes, they are actually two main stems of a single basal root system and trunk. Both tree 13 and tree 14 originated as stump sprouts off of an older tree that was knocked down over 300 years ago. Attempts

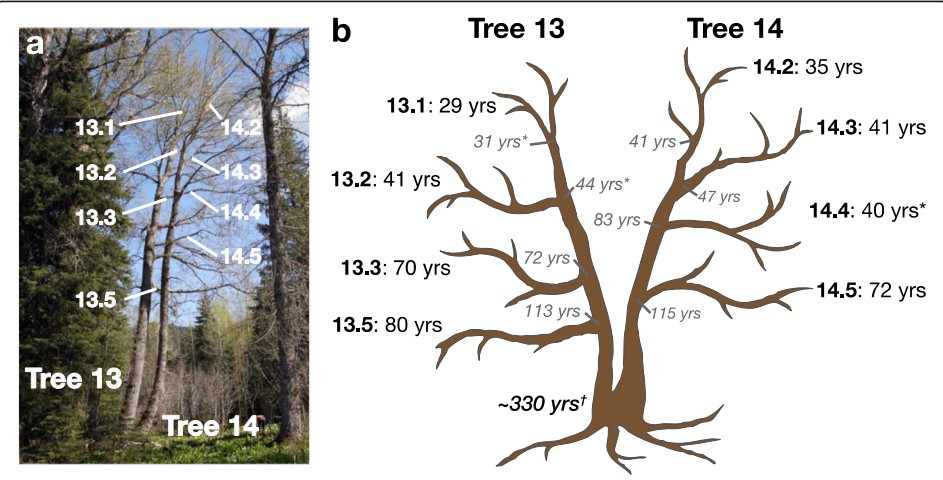

**Fig. 1** Photograph and schematic drawing of tree 13 and tree 14. This wild *P. trichocarpa*, located near Mt. Hood, Oregon, experienced a decapitation event ~ 300 years ago. Tree 14 re-sprouted from the stump and ~ 80–100 years later tree 13 re-sprouted. **a** Leaf samples were collected from the labeled terminal branches. **b** Age was estimated for both the end of the branch (black font) and where it meets the main stem (gray italics). Ages with * indicate age was estimated using diameter; all other estimates were from core samples. Leaf samples of each branch was used to create genomic sequencing libraries, PacBio libraries, whole-genome bisulfite sequencing libraries, and mRNA-sequencing libraries

to determine the total age were unsuccessful. However, statistical estimates based on molecular-clock arguments and a regression analysis of diameter to age suggest that the tree is approximately 330 years old (Shayary et al. 2019, co-submission).

Leaf samples were collected from eight age-estimated branches for multi-omics analysis for tree 13 and tree 14. The oldest branch of tree 14 (branch 14.5) was used for genome assembly of *Populus trichocarpa* var. *Stettler*. Genome resequencing was performed for all branches to explore intra- and inter-tree genetic variation. PacBio, MethylC-seq, and mRNA-seq libraries were constructed for the branches of tree 13 and tree 14 to explore structural, methylation, and transcriptional variation, respectively.

## Genome assembly and annotation of *Populus trichocarpa* var. *Stettler*

We sequenced the *P. trichocarpa* var. *Stettler* using a whole-genome shotgun sequencing strategy and standard sequencing protocols. Sequencing reads were collected using Illumina and PacBio. The current release is based on PacBio reads (average read length of 10,477 bp, average depth of 118.58×) assembled using the MECAT CANU v.1.4 assembler [39] and subsequently polished using QUIVER [40]. A set of 64,840 unique, non-repetitive, non-overlapping 1.0-kb sequences were identified in the version 4.0 *P. trichocarpa* var. *Nisqually* assembly and were used to assemble the chromosomes. The version 1 *Stettler* release contains 392.3 Mb of sequence with a contig N50 of 7.5 Mb and 99.8% of the assembled sequence captured in the chromosomes. Additionally, ~ 232.2 Mb of alternative haplotypes were identified. Completeness of the final assembly was assessed using 35,172 annotated genes from the version 4.0 *P. trichocarpa* var. *Nisqually* release (jgi.doe.gov). A total of 34,327 (97.72%) aligned to the primary *Stettler* assembly.

The annotation was performed using ~ 1.4 billion pairs of $2 \times 150$ stranded paired-end Illumina RNA-seq GeneAtlas *P. trichocarpa* var. *Nisqually* reads, ~ 1.2 billion pairs of $2 \times 100$ paired-end Illumina RNA-seq *P. trichocarpa* var. *Nisqually* reads from Dr. Pankaj Jaiswal, and ~ 430 million pairs of $2 \times 75$ stranded paired-end Illumina var. *Stettler* reads using PERTRAN (as described in [41]) on the *P. trichocarpa* var. *Stettler* genome. About ~ 3 million PacBio Iso-Seq circular consensus sequences were corrected and collapsed by a genome-guided correction pipeline on the *P. trichocarpa* var. *Stettler* genome to obtain ~ 0.5 million putative full-length transcripts. We annotated 34,700 protein-coding genes and 17,314 alternative splices for the final annotation. Because of the extensive resources included in the annotation, 32,330 genes had full-length transcript support.

## Identification and rate of somatic genetic variants

Leaf samples from the five trees were sequenced to an average depth of ~ 87× (~ 60–164×) using Illumina HiSeq. Roughly 88% of the high-quality reads map to the genome and about 98.6% of the genome is covered by at least one read, and genome coverage (~ 8–500×) used for SNP calling was about 97%. The initial number of SNPs per tree (mutation on any branch) varied between 44,000 and 152,000, which is populated with many false positives due to coverage, sequencing, alignment errors, etc. Applying an additional filter requiring > 20× coverage per position and requiring coverage in all branches reduced the total amount genome space queried to ~ 40 Mb. Furthermore, since most of the genome (99.9%) is homozygous at every base pair, a somatic mutation will almost always result in a change from a homozygous to heterozygous site. Restricting the analysis to sites that change from homozygous to heterozygous, we identified 91 high-confidence SNPs in tree 13 and 95 high-confidence SNPs in tree 14 (Additional file 2: Tables S1–2).

Over two thirds of the SNPs in tree 13 and tree 14 were transition mutations, with C-G to T-A mutations accounting for over 54% of the SNPs (Fig. 2a). Of the transversion mutations, C-G to G-C was the least common (3.8%) whereas C-G to A-T was most common (10%). Nearly half of the SNPs (46%) occurred in transposable elements and about 10% occur in promoter regions (Fig. 2b and Additional file 2: Tables S1-S2). SNPs are significantly enriched in TEs and depleted in mRNA regions genome-wide (chi-square, df = 3, $P < 0.001$). For TEs, SNPs have a 0.66 fold-enrichment in tree 13 and 0.85 fold-enrichment in tree 14 (Additional file 2: Table S3a). Examining TE classes further, SNPs are enriched in SINE elements are depleted in Gypsy and Helitron elements (Additional file 2: Table S3b).

To obtain an estimate of the rate of somatic point mutations from these SNP calls, we developed *mutSOMA* (https://github.com/jlab-code/mutSOMA), a phylogeny-based inference method that fully incorporates knowledge of the age-dated branching topology of the tree (see "Methods" and Additional file 3: Supplementary Text). Using this approach, we find that the somatic point mutation rate in poplar is $1.33 \times 10^{-10}$ (95% CI $1.53 \times 10^{-11}$–$4.18 \times 10^{-10}$) per base per haploid genome per year (Additional file 2: Table S4). Generation time can refer to two measurements—time from seed to production of first seeds and the organism's lifespan. In annual plants, these values can be considered the same; however, this is not the case for perennials. Assuming 15 years from seed to

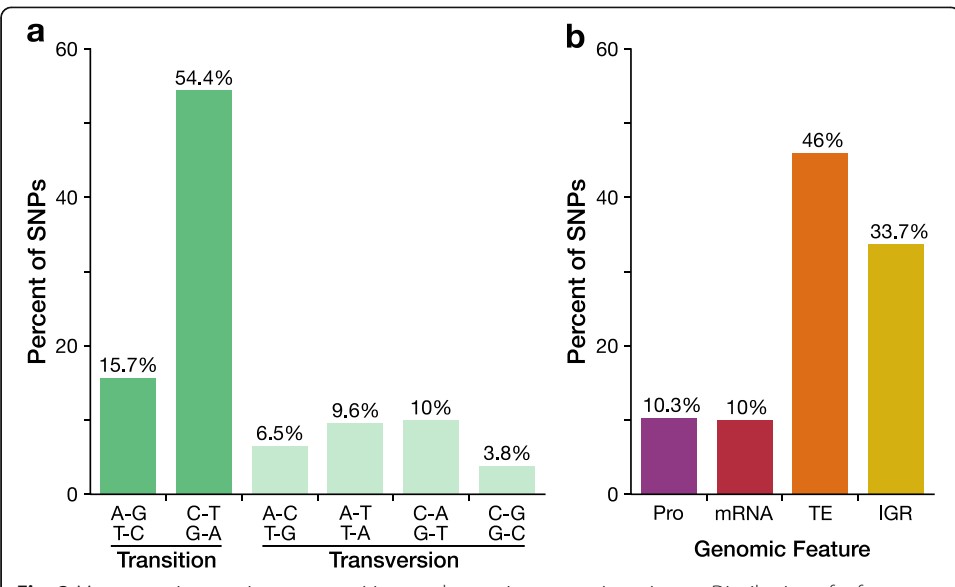

**Fig. 2** Most somatic mutations are transitions and occur in non-genic regions. **a** Distribution of reference to alternative allele observed in the high-confidence SNPs identified in tree 13 and tree 14. **b** Distribution of high-confidence SNPs separated by the genomic feature. See Additional file 2: Table S3 for per-tree distributions. Abbreviations: Pro, promoter (2 kb upstream of TSS); TE, transposable elements and repeats; and IGR, intergenic regions

production of first seeds [42], the poplar seed-to-seed generation mutation rate would be approximately $1.99 \times 10^{-9}$. This is slightly lower than the per-generation (seed-to-seed) mutation rate observed in the annual *A. thaliana* ($7 \times 10^{-9}$) [16]. Next looking at the life-span per-generation rate and assuming a maximum age of 200 years [43], the lifespan per-generation rate is $2.66 \times 10^{-8}$. This estimate is slightly lower than the per-generation somatic mutation rate recently reported in oak ($4.2–5.8 \times 10^{-8}$) [14].

To analyze structural variants (SV) between haplotypes and somatic SV muta-tions, PacBio libraries were generated for the eight branches from tree 13 and tree 14 (Fig. 1). For each branch, four PacBio cells were sequenced generating an aver-age output of 3.05 million reads and 28.3 Gb per branch (Additional file 2: Table S5). After aligning the PacBio output to the *P. trichocarpa* var. *Stettler* genome, calling SVs larger than 20 bp, and filtering, we identified ~ 10,466 deletions, ~ 6702 insertions, 645 duplications, and three inversions between the reference *Stettler* haplotype and the alternative haplotype (Additional file 2: Table S6). Upon manual inspection of read mapping for a representative subset of SVs, 72.6% of SVs have strong support where multiple aligned reads support the SV type and size (Add-itional file 2: Table S7). Deletions and duplications are significantly enriched in tandem repeat sequence and depleted in genic sequence (Kolmogorov-Smirnov two-sample test, *P* value $< 2.2 \times 10^{-16}$). Furthermore, deletions generally have less genic sequence and more tandem repeat sequence than do duplications (Additional file 1: Fig. S3). Several of the detected SVs are large, with 11 deletions and five du-plications greater than 50 kb (Additional file 2: Table S6) with genic sequence con-tent ranging from 0.0 to 23.7%. Comparisons of the branches from tree 13 and tree 14 did not identify instances of somatic SV mutation.

### Identification and rate of somatic epigenetic variants

To explore somatic epigenetic variation associated with changes in DNA methylation, we generated whole-genome bisulfite sequencing libraries from the branch tips of tree 13 and tree 14 (Fig. 1). The average genome coverage for the samples was ~ 41.1×, and sequence summary statistics are located in Additional file 2: Table S8. Genome-wide methylation levels were similar across all samples with 36.61% mCG, 19.02% mCHG, and 2.07% mCHH% (Additional file 1: Fig. S4) [44], indicating that global methylation levels are relatively stable across branches. Nonetheless, we observed significant age-dependent DNA methylation divergence between branches in CG and CHG contexts, both at the level of individual cytosines as well as at the level of regions, i.e., clusters of cytosines (Fig. 3a-b, Additional file 1: Fig. S5, and Additional file 2: Table S9). These age-dependent divergence patterns indicate that spontaneous methylation changes (i.e., epimutations) are cumulative across somatic development and thus point to a shared meristematic origin (Shahryary et al. 2019, co-submission).

To obtain an estimate of somatic epimutation rates, we applied *AlphaBeta* (Shahryary et al. 2019, co-submission). The method builds on our previous approach for estimating "germline"-epimutation in mutation accumulation (MA) lines except here we treat the tree branching topology as an intraorganismal phylogeny and model mitotic instead of meiotic inheritance. Focusing first on cytosine-level epimutations, we estimated that at the genome-wide scale spontaneous methylation gains in contexts CG and CHG occur at a rate of $1.8 \times 10^{-6}$ and $3.3 \times 10^{-7}$ per site per haploid genome per year, respectively, whereas spontaneous methylation losses in these two sequence contexts occur at a rate of $5.8 \times 10^{-6}$ and $4.1 \times 10^{-6}$ per site per haploid genome per year. Similar rate estimates were obtained in a replication experiment (Additional file 1: Fig. S6). Cytosines in CHH context could not be shown to significantly accumulate epimutations (Additional file 2: Table S9). Based on these estimates, we extrapolate that the *seed-to-seed* per-generation epimutation rate in poplar is about $10^{-5}$ and the *lifespan* per-generation rate is $10^{-4}$. Remarkably, these estimates are very similar to those reported in *A. thaliana* MA lines where the average CG and CHG rates are about $3.6 \times 10^{-4}$ and $3.1 \times 10^{-5}$, respectively (Shahryary et al. 2019, co-submission). The observation that two species with such different life history traits and genome architecture display very similar per-generation mutation and epimutation rates suggests that the rates themselves are subject to strong evolutionary constraints.

In addition to global epimutation rates, we also estimated rates for different genomic features (mRNA, promoters, intergenic, TEs). This analysis revealed highly significant rate differences in the CG and CHG context between genomic features, with mRNAs showing the highest and TEs the lowest combined rates (Fig. 3c–j). Interestingly, the ordering of the magnitude of the mRNA, promoter, and intergenic rates is similar to that previously observed in *A. thaliana* MA lines [34]. The differences in rates at local genomic features likely reflect the distinct DNA methylation pathways that function on these sequences (RNA-directed DNA methylation, CHROMOMETHYLASE3, CHROMOMETHYLASE2, DNA METHYLTRANSFERASE1, etc.). For example, the high rate of epimutation losses in mRNA relative to other features (Fig. 3g, h) could reflect the activity of CMT3-mediated gene body DNA methylation [45, 46]. The observation that the epimutation rates of these features are consistent between *A. thaliana* MA lines (> 60 generations) and this long-lived perennial (within a single generation) seems to

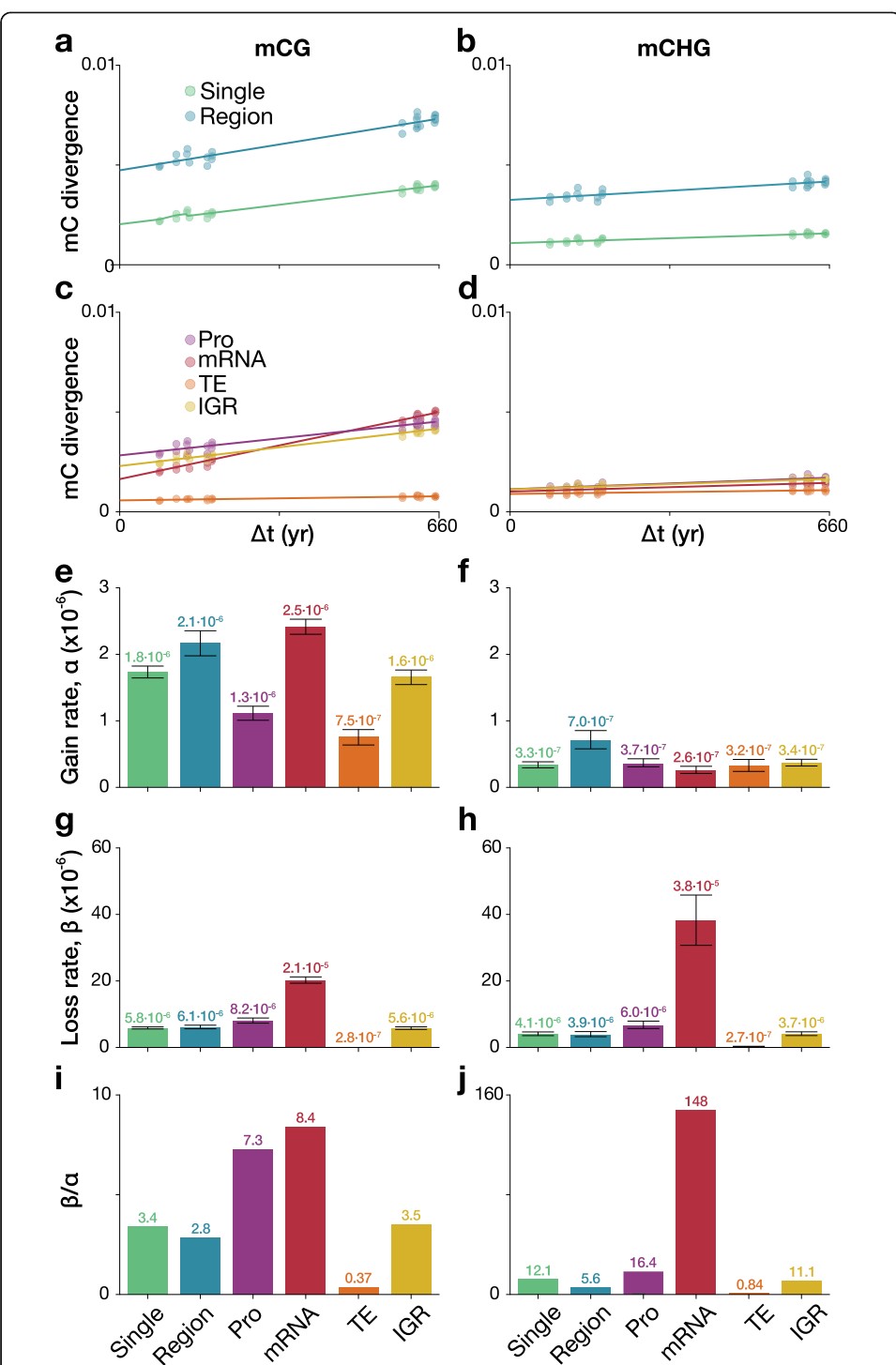

**Fig. 3** Somatic epimutation rates for single sites, regions, and by genomic feature. mCG (**a**) and mCHG (**b**) divergence by branch time divergence for single sites and regions; mCG (**c**) and mCHG (**d**) divergence by branch time divergence for genomic features Pro (promoter; 2 kb upstream of TSS), mRNA, TE (transposable elements), and IGR (intergenic regions). The dots show the individual observed divergences, whereas the line represents the fit of the data to the model. Estimated mCG (**e**) and mCHG (**f**) gain rates by feature. Estimated mCG (**g**) and mCHG (**h**) loss rates by feature. Ratio of mCG (**i**) and mCHG (**j**) loss to gain by feature. An F-test was used comparing the neutral model vs null model (Supplementary Text). See Additional file 2: Table S9 for *P* values. Error bars represent bootstrapped 95% confidence intervals of the estimates. Abbreviations: Pro, promoter; TE, transposable elements and repeats; and IGR, intergenic regions

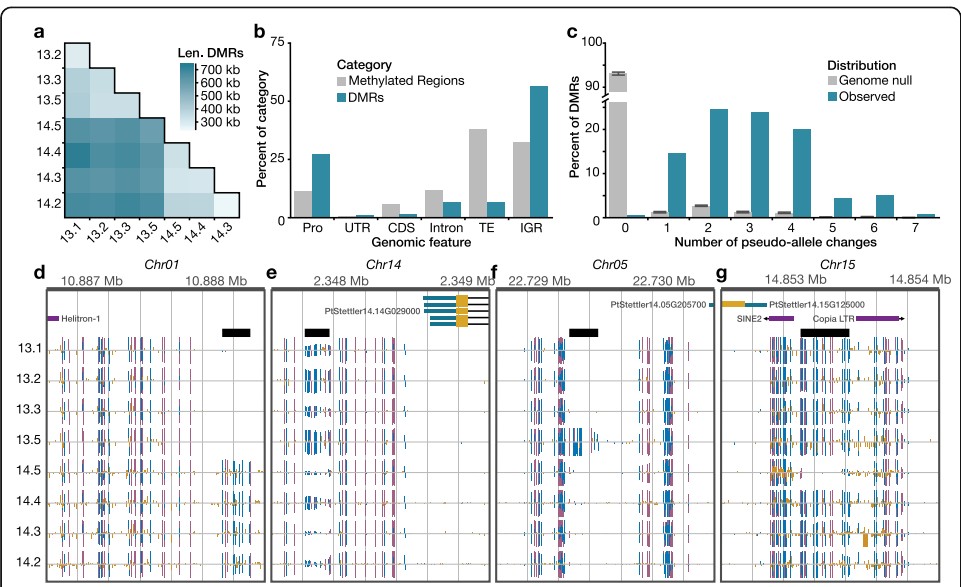

**Fig. 4** Identification and quantification of somatic stability of differentially methylated regions. **a** Divergence of differentially methylated regions corresponds to divergence in age. The darker color indicates combined length of the pairwise DMRs. **b** The genome-wide distribution of identified DMRs in genomic features. Abbreviations: TE, transposable elements and repeats; IGR, intergenic region; Pro, promoter region (2 kb upstream of TSS); UTR, untranslated regions; CDS, coding sequence. Methylated regions were identified in as regions methylated in at least one sample. **c** There are significantly more pseudo-allele changes between the branches at DMRs (blue) compared to the genome-wide null (Wilcoxon rank sum, one-sided, *P* value < $2 \times 10^{-16}$). Gray bars are the genome-wide null as mean +/− std. dev. across 10 simulations. **d** Browser screenshot of a tree specific DMR where all branches in tree 13 are homozygous unmethylated and all branches of tree 14 are homozygous methylated. **e** Browser screenshot of a highly variable DMR where the pseudo-allele state changes between each branch. **f** Browser screenshot of a single gain DMR where all branches except 13.5 are homozygous unmethylated and 13.5 gains methylation. **g** Browser screenshot of a single loss DMR where all branches except 14.5 are homozygous methylated and 14.5 has lost methylation. For **d**–**g**, gene models and transposable elements are shown at the top and methylome tracks are below. Vertical bars indicate methylation at the position, where height corresponds to level and color is context, red for CG, blue for CHG, and yellow for CHH. DMR is indicated by thick black horizontal line

imply that epimutations are not a result of biased reinforcement of DNA methylation during sexual reproduction or environment/genetic variation, but instead a feature of DNA methylation maintenance through mitotic cell divisions.

### Assessment of spontaneous differentially methylated regions

Differentially methylated regions are functionally more relevant than individual cytosine-level changes, as in certain cases they are linked to differential gene expression and phenotypic variation [27–29, 47, 48]. To explore the extent of differentially methylated regions (DMRs) that spontaneously arise in these trees, we searched for all pairwise DMRs between all branches. In total, we identified 10,909 DMRs that possessed changes in all sequence contexts (CG, CHG, and CHH-C-DMRs). Together, they constitute approximately 1.69 Mb of the total 167.4 Mb (~ 1%) of methylated sequences in the *Stettler* genome and they reveal age-dependent accumulation (Fig. 4a). Most DMRs occur in intergenic regions (56.7%), but a significant enrichment of DMRs was detected within 2 kb from the transcriptional start site of genes compared to methylated regions

as a whole (Fig. 4b) (Fisher's exact test, one-sided, *P* value < 0.001). There is no significant enrichment of gene function for DMRs within promoters.

Given the heterozygous nature of wild *P. trichocarpa*, we explored allelic methylation changes. After filtering for sufficient coverage and methylation change, we assigned the pseudo-allele state of each branch at 4488 DMRs. Possible states were homozygous unmethylated, heterozygous, and homozygous methylated. In each sample, 43.0% of DMRs, on average, were categorized as homozygous methylated (Additional file 1: Fig. S7). Interestingly, the youngest branches, 13.1 and 14.1 have about 10% more homozygous methylated pseudo-alleles than the other branches (51.1% vs 41.7%). Next, we looked at the number of changes of pseudo-allele states. This is expected as DMRs were identified as having different methylation levels in the samples. On average, there are 3.02 state changes for each DMR with 94.4% of DMRs having one to five state changes (Fig. 4c). These data suggest that many of these regions are metastable, a common feature of epimutations in plants [27–31, 33].

An example of a region with one state change are the tree specific DMRs (Fig. 4d). In these regions, all branches of one tree are homozygous unmethylated and all branches of the other tree are homozygous methylated. This suggesting the methylation state change occurred shortly after the trees separated and remained stable throughout subsequent mitotic divisions. In contrast, we also identified highly variable regions with seven state changes, a change between each branch (Fig. 4e). Of the regions with two state changes, 150 have branch-specific state changes. For example, in Fig. 4f, branches 13.1 to 13.3 are homozygous unmethylated, then it changes to homozygous methylated for branch 13.5, and changes again to homozygous unmethylated for branches 14.5–14.2. Similarly, in Fig. 4g, all branches except 14.5 are homozygous methylated and 14.5 has spontaneously lost methylation.

Analogous to our epimutation rate estimation of individual sites, we used the identified C-DMRs (differential methylation in all cytosine sequence contexts) as a basis to obtain estimates of the rate at which such region-level changes occur. To do this, we separated the remaining genomic space into control regions ("non-DMR") with the same size distribution as observed for C-DMRs and used the methylation status of all (non-) DMR as input for epimutation analysis. We found that only 17% of single CG and CHG epimutations were located inside the identified C-DMRs. Despite this, our analysis shows that region-level epimutation rates are similar to epimutation rates of single cytosines (Fig. 3 and Additional file 2: Table S8). This observation can be explained by the fact that the total number of regions is also much smaller than the number of individual cytosines.

### Functional consequences of differential methylation on gene expression

To assess if age-related cytosine methylation changes have functional consequences, we performed mRNA-seq with three biological replicates for each branch of trees 13 and 14. On average, each library had over ~ 55 million reads and 96.8% mapping to the *P. trichocarpa* var. *Stettler* genome (Additional file 2: Table S10). We used DESeq2 to identify differentially expressed genes (DEGs) pairwise between branches [49] and identified a total of 2937 genes. The *P. trichocarpa* var. *Stettler* genome has 34,700

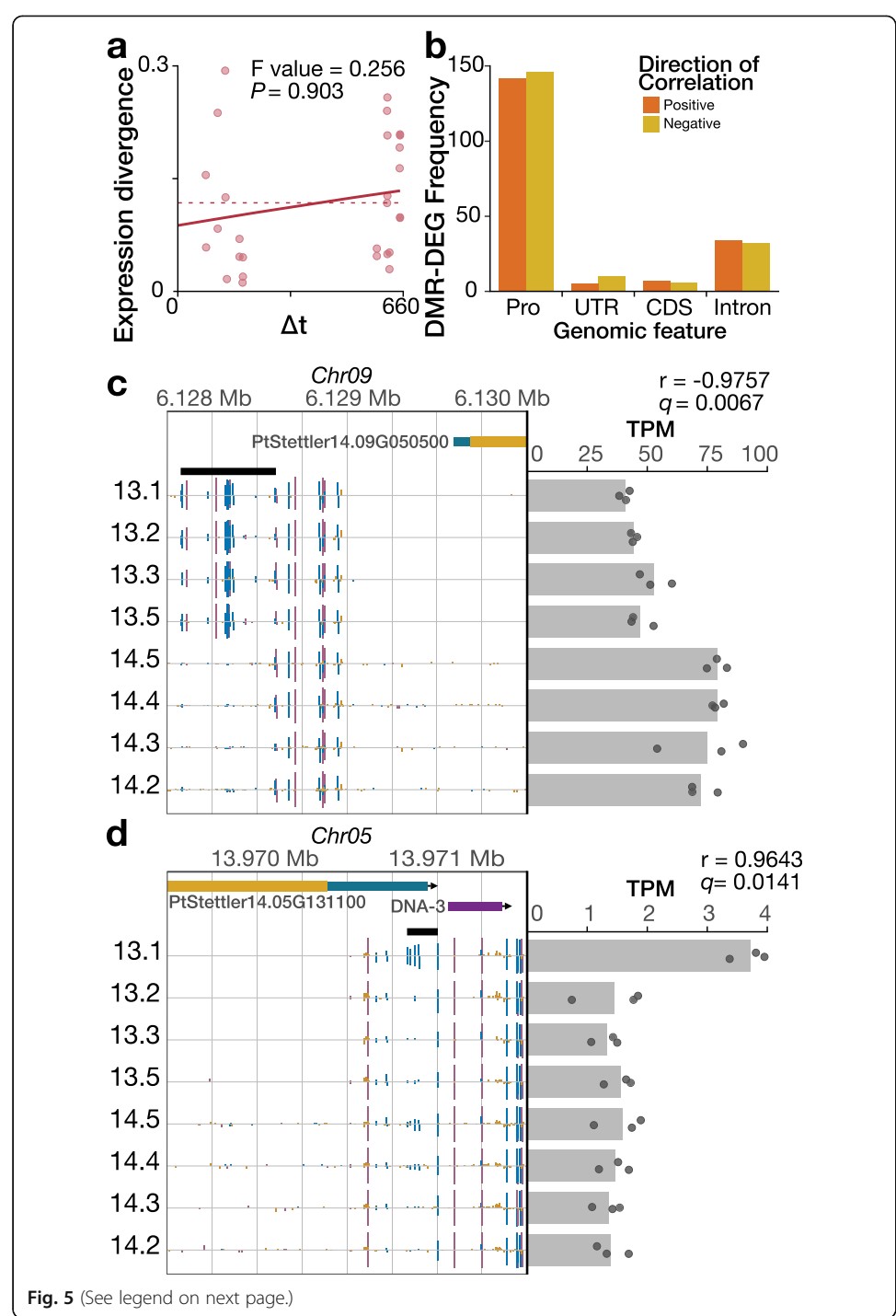

**Fig. 5** (See legend on next page.)

(See figure on previous page.)

**Fig. 5** Gene expression is largely independent from divergence age and nearby cytosine methylation except in a few examples. **a** Gene expression divergence is not significantly associated with divergence age. The dots show the individual observed divergences, whereas the line represents the fit of the data to the model. **b** Distribution of positive and negative correlations for differentially expressed genes and overlapping/nearby DMRs. Positive correlation occurs when the higher methylation level is associated with higher gene expression among the samples. **c** A significantly negatively correlated, tree-specific DMR and DEG where the DMR occurs in the promoter region of the gene (Pearson's correlation test, two-sided, $N = 8$, adjusted $P$ value = 0.0067). The higher methylation levels in the DMR for tree 13 branches are associated with lower gene expression. **d** A significantly positively correlated, single gain DMR and DEG where the DMR occurs in the 5′ untranslated region of the gene (Pearson's correlation test, two-sided, $N = 8$, adjusted $P = 0.0141$). The higher methylation level in the DMR for branch 13.1 is associated with greater gene expression. For **c** and **d**, gene expression, as transcripts per million (TPM), is represented as points for the individual replicates and as bar for mean among replicates. In the genome browser view, gene models and transposable elements are shown at the top and methylome tracks are below. Vertical bars indicate methylation at the position, where height corresponds to level and color is context, red for CG, blue for CHG, and yellow for CHH. DMR is indicated by thick black horizontal line

annotated genes, so this differential expression gene set is 8.46% of all genes and 10.5% of expressed genes.

Since the somatic accumulation of spontaneous methylation changes could affect gene expression, we asked if transcriptional divergence also increases as a function of tree age. We found that in contrast to somatic mutations and epimutations, the divergence between leaf transcriptomes is much more heterogeneous and displays only a weak and non-significant accumulation trend (Fig. 5a). Linear regression analysis showed no statistically significant relationship between transcription and methylation over time ($P$ value = 0.1293). These results show that divergence in DNA methylation is not accompanied by transcriptional divergence during tree aging, probably as a result of gene expression being much more dynamic and responsive to current environments.

However, this global analysis does not rule out that DNA methylation changes at specific individual loci can have transcriptional consequences. To explore this in more detail, we analyzed DMRs proximal to DEGs and correlated the methylation level of the DMR with the expression level of the gene. The correlation is positive when a higher methylation level in the DMR is associated with higher expression of the gene and negative when higher methylation is associated with lower expression of the gene. Regardless of where the DMR was located relative to the gene, we observed positive DMR-DEG correlations and negative DMR-DEG correlations. There was no bias for direction of correlation and genomic feature type (Fig. 5b). The negative correlations represent cases where DNA methylation is blocking cis-regulatory elements, whereas the positive correlations could represent cases where increased accessibility to transcriptional machinery of the gene leads to greater RdDM activity as observed previously [50].

We further focused on four specific examples where DEG-DMR correlations were statistically significant (Additional file 1: Fig. S8). Of these four, three of the DMRs occurred within 2 kb upstream of the transcription start site, and they have strong negative correlations (Fig. 5c). The DMR located in the untranslated region of a gene encoding a mitochondrial oxoglutarate/malate carrier protein was positively correlated with gene expression (Fig. 5d), although it remains unclear if this relationship is causal.

Taken together, our transcriptome analysis indicates that gene expression changes in this poplar tree are largely independent of methylation at both the global and local

scale except for a few rare examples. This observation is at least partly consistent with our model-based analyses, which suggest that somatic epimutations in this tree accumulate neutrally (Shahryary et al. 2019, co-submission).

## Discussion

Using a multi-omics approach, we were able to calculate the rates of somatic mutations and epimutations in the long-lived perennial tree *P. trichocarpa*. Consistent with the per-unit-time hypothesis, we find that the per-year genetic and epigenetic mutation rates in poplar are lower than in *A. thaliana*, which is remarkable considering that the former experienced hundreds of years of variable environmental conditions. This observation supports the view that long-lived perennials may limit the number of meristematic cell divisions during their lifetime and that they have evolved mechanisms to protect these cell types from the persistent influence of environmental mutagens, such as UV radiation. Interestingly, in contrast to the observed differences in *per-year* mutation and epimutation rates, our analysis reveals strong similarities in the *per-generation* rates between these two species. This close similarity further suggests that the per-generation rates of spontaneous genetic and epigenetic changes are under strong evolutionary constraint, although it remains unclear from our experimental design how many of these (epi) mutations will be successfully transferred to the next generation.

The results presented here are most certainly an underestimate of the actual rate. This may be a result of the sampling biased used in this study, as we were only able to sample surviving branches and identify mutations that occurred early enough that they are present in the majority of the cells sampled in the tissues profiled. Perhaps variable environmental conditions lower the epimutation rate by keeping the cells in sync, thus few differences can be observed. Alternatively, meristematic cells that give rise to the sampled tissues have highly reinforced and well-maintained DNA methylomes similar to observations in embryonic tissue [51–55]. Either scenario would imply that most of the identified epimutations are spontaneous in nature. Although the rate is different, the ordering in feature-specific epimutation rates is the same between poplar and *A. thaliana*, suggesting that this is a general pattern in plant genomes, which likely is derived from maintenance of DNA methylation through mitotic cell divisions.

The biological significance of the majority of newly formed epimutations is unclear at this time, although most are likely neutral. It is noteworthy that some of the identified epimutations are associated with expression variation; however, this was a rare occurrence in this study. These results also reflect the rarity at which epimutations linked to morphological variation are found in the laboratory and/or field [27–31, 33]. One emerging hypothesis is that the majority of epimutations in angiosperms are byproducts of maintenance of DNA methylation associated with heterochromatin and that certain loci are more or less susceptible than others [45, 46, 56]. This link to maintenance processes is one commonality between the accumulation of genetic and epigenetic changes that could explain why their rates are fairly similar on a per cell division timescale.

## Conclusion

Taken together, our study provides unprecedented insights into the origin of nucleotide, epigenetic, and functional variation in the long-lived perennial plant.

## Methods

### Sample collection and age estimation

The trees used in this study were located at Hood River Ranger District [Horse Thief Meadows area], Mt. Hood National Forest, 0.6 mi south of Nottingham Campground off OR-35 at unmarked parking area, 500′ west of East Fork Trail #650 across river, ca. 45.355313, – 121.574284 (Additional file 1: Fig. S1).

Cores were originally collected from the main stem and five branches from each of five trees in April 2015 at breast height (~ 1.5 m) for standing tree age using a stainless-steel increment borer (5 mm in diameter and up to 28 cm in length). Cores were mounted on grooved wood trim, dried at room temperature, sanded, and stained with 1% phloroglucinol following the manufacturer's instructions (https://www.forestry-suppliers.com/Documents/1568_msds.pdf). Annual growth rings were counted to estimate age. For cores for which accurate estimates could not be made from the 2015 collection, additional collections were made in spring 2016. However, due to difficulty in collecting by climbing, many of the cores did not reach the center of the stem or branches (pith) and/or the samples suffered from heart rot. Combined with the difficulty in demarcating rings in porous woods such as poplar *Populus* [57, 58], accurate measures of tree age or branch age were challenging (Additional file 1: Fig. S2).

Simultaneously with stem coring, leaf samples were collected from the tips of each of the branches from the selected five trees. Branches 9.1, 9.5, 13.4, 14.1, 15.1, and 15.5 were too damaged to determine reasonable age estimates and were removed from analysis. Branch 14.4 and the stems of 13.1 and 13.2 were estimated by simply regressing the diameter of all branches and stems that could be aged by coring.

### Nuclei prep for DNA extraction

Poplar leaves that had been kept frozen at – 80 °C were gently ground with liquid nitrogen and incubated with NIB buffer (10 mM Tris-HCL, PH8.0, 10 mM EDTA PH8.0, 100 mM KCL, 0.5 M sucrose, 4 mM spermidine, 1 mM spermine) on ice for 15 min. After filtration through miracloth, Triton x-100 (Sigma) was added to tubes at a 1:20 ratio, placed on ice for 15 min, and centrifuged to collect nuclei. Nuclei were washed with NIB buffer (containing Triton x-100) and re-suspended in a small amount of NIB buffer (containing Triton x-100) then the volume of each tube was brought to 40 ml and centrifuged again. After careful removal of all liquid, 10 ml of Qiagen G2 buffer was added followed by gentle re-suspension of nuclei; then 30 ml G2 buffer with RNase A (to final concentration of 50 mg/ml) was added. Tubes were incubated at 37 °C for 30 min. Proteinase K (Invitrogen), 30 mg, was added and tubes were incubated at 50 °C for 2 h followed by centrifugation for 15 min at 8000 rpm, at 4 °C, and the liquid gently poured to a new tube. After gentle extraction with chloroform to isoamyl alcohol (24:1), then centrifugation and transfer of the top phase to a fresh tube, HMW DNA was precipitated by addition of 2/3 volume of iso-propanol and re-centrifugation to collect the DNA. After DNA was washed with 70% ethanol, it was air dried for 20 min and dissolved thoroughly in 1× TE.

### Whole-genome sequencing

We sequenced *Populus trichocarpa* var. *Stettler* using a whole-genome shotgun sequencing strategy and standard sequencing protocols. Sequencing reads were collected using Illumina and PacBio. Both the Illumina and PacBio reads were sequenced at the Department of Energy (DOE) Joint Genome Institute (JGI) in Walnut Creek, California, and the HudsonAlpha Institute in Huntsville, Alabama. Illumina reads were sequenced using the Illumina HISeq platform, while the PacBio reads were sequenced using the RS platform. One 400-bp insert $2 \times 150$ Illumina fragment library was obtained for a total of $\sim 349\times$ coverage (Additional file 2: Table S11). Prior to assembly, all Illumina reads were screened for mitochondria, chloroplast, and phix contamination. Reads composed of > 95% simple sequence were removed. Illumina reads less than 75 bp after trimming for adapter and quality ($q < 20$) were removed. The final Illumina read set consists of 906,280,916 reads for a total of $\sim 349\times$ of high-quality Illumina bases. For the PacBio sequencing, a total of 69 chips (P6C4 chemistry) were sequenced with a total yield of 59.29 Gb (118.58×) with 56.2 Gb > 5 kb (Additional file 2: Table S12), and post error correction a total of 37.3 Gb (53.4×) was used in the assembly.

### Genome assembly and construction of pseudomolecule chromosomes

The current release is version 1.0 release began by assembling the 37.3 Gb corrected PacBio reads (53.4× sequence coverage) using the MECAT CANU v.1.4 assembler [39] and subsequently polished using QUIVER v.2.3.3 [40]. This produced 3693 scaffolds (3693 contigs), with a scaffold N50 of 1.9 Mb, 955 scaffolds larger than 100 kb, and a total genome size of 693.8 Mb (Additional file 2: Table S13). Alternative haplotypes were identified in the initial assembly using an in-house Python pipeline, resulting in 2972 contigs (232.3 Mb) being labeled as alternative haplotypes, leaving 745 contigs (461.5 Mb) in the single haplotype assembly. A set of 64,840 unique, non-repetitive, non-overlapping 1.0-kb syntenic sequences from version 4.0 *P. trichocarpa* var. *Nisqually* assembly aligned to the MECAT CANU v.1.4 assembly and used to identify misjoins in the *P. trichocarpa* var. *Stettler* assembly. A total of 22 misjoins were identified and broken. Scaffolds were then oriented, ordered, and joined together into 19 chromosomes. In the syntenic marker FASTA file, each record identifier carried information pertaining to the *Nisqually* chromosome where the sequence was extracted, as well as the position in the chromosome. These markers, along with the annotated primary transcripts from *Nisqually*, were aligned to the Poplar var. 14.5 assembly using BLAT. The chromosome/position information was used to identify misjoins in the assembly. Once the misjoins were corrected, the scaffolds were ordered and oriented using the positional information contained in the syntenic markers/genes. A total of 117 joins were made during this process, and the chromosome joins were padded with 10,000 Ns [59]. Small adjacent alternative haplotypes were identified on the joined contig set. Alternative haplotype (Althap) regions were collapsed using the longest common substring between the two haplotypes. A total of 14 adjacent alternative haplotypes were collapsed.

The resulting assembly was then screened for contamination. Homozygous single nucleotide polymorphisms (SNPs) and insertion/deletions (InDels) were corrected in the release sequence using $\sim 100\times$ of Illumina reads ($2 \times 150$, 400-bp insert) by aligning the

reads using bwa-0.7.17 mem [60] and identifying homozygous SNPs and InDels with the GATK v3.6's UnifiedGenotyper tool [61]. A total of 206 homozygous SNPs and 11,220 homozygous InDels were corrected in the release. Heterozygous SNP/indel phasing errors were corrected in the consensus using the 118.58× raw PacBio data [59]. A total of 66,124 (1.98%) of the heterozygous SNP/InDels were corrected. The final version 1.0 improved release contains 391.2 Mb of sequence, consisting of 25 scaffolds (128 contigs) with a contig N50 of 7.5 Mb and a total of 99.8% of assembled bases in chromosomes. Plots of the *Nisqually* marker placements for the 19 chromosomes are shown in Additional file 1: Fig. S9.

## Genome annotation

Transcript assemblies were made from ~ 1.4 billion pairs of $2 \times 150$ stranded paired-end Illumina RNA-seq GeneAtlas *P. trichocarpa* var. *Nisqually* reads, ~ 1.2 billion pairs of $2 \times 100$ paired-end Illumina RNA-seq *P. trichocarpa* var. *Nisqually* reads from Dr. Pankaj Jaiswal, and ~ 430 M pairs of $2 \times 75$ stranded paired-end Illumina var. *Stettler* reads using PERTRAN [41] on *P. trichocarpa* var. *Stettler* genome. About ~ 3 M PacBio Iso-Seq circular consensus sequences were corrected and collapsed by genome-guided correction pipeline on *P. trichocarpa* var. *Stettler* genome to obtain ~ 0.5 million putative full-length transcripts. A total of 293,637 transcript assemblies were constructed using PASA [62] from RNA-seq transcript assemblies above. Loci were determined by transcript assembly alignments and/or EXONERATE alignments of proteins from *A. thaliana*, soybean, peach, Kitaake rice, *Setaria viridis*, tomato, cassava, grape, and Swiss-Prot proteomes to repeat-soft-masked *P. trichocarpa* var. *Stettler* genome using RepeatMasker [63] with up to 2-kb extension on both ends unless extending into another locus on the same strand. Gene models were predicted by homology-based predictors, FGENESH+ [64] FGENESH_EST (similar to FGENESH+, EST as splice site and intron input instead of protein/translated ORF), EXONERATE [65], PASA assembly ORFs (in-house homology constrained ORF finder), and from AUGUSTUS via BRAKER1 [66]. The best scored predictions for each locus are selected using multiple positive factors including EST and protein support, and one negative factor: overlap with repeats. The selected gene predictions were improved by PASA. Improvement includes adding UTRs, splicing correction, and adding alternative transcripts. PASA-improved gene model proteins were subject to protein homology analysis to the abovementioned proteomes to obtain Cscore and protein coverage. Cscore is a protein BLASTP score ratio to MBH (mutual best hit) BLASTP score, and protein coverage is the highest percentage of protein aligned to the best of homologs. PASA-improved transcripts were selected based on Cscore, protein coverage, EST coverage, and its CDS overlapping with repeats. The transcripts were selected if its Cscore is larger than or equal to 0.5 and protein coverage larger than or equal to 0.5, or it has EST coverage, but its CDS overlapping with repeats is less than 20%. For gene models whose CDS overlaps with repeats for more than 20%, its Cscore must be at least 0.9 and homology coverage at least 70% to be selected. The selected gene models were subject to Pfam analysis and gene models whose protein is more than 30% in Pfam TE domains were removed and weak gene models.

Incomplete gene models, low homology supported without fully transcriptome supported gene models and short single exon (< 300-bp CDS) without protein domain nor good expression gene models, were manually filtered out.

### SNP calling methods

Illumina HiSeq2500 paired-end (2 × 150) reads were mapped to the reference genome using bwa-mem [60]. Picard toolkit was used to sort and index the bam files. GATK [61] was used further to align regions around InDels. Samtools v1.9 [67] was used to create a multi-sample mpileup for each tree independently. Preliminary SNPs were called using Varscan v2.4.0 [68] with a minimum coverage of 21.

At these SNPs, for each branch, we calculated the conditional probability of each potential genotype (RR, RA, AA) given the read counts of each allele, following SeqEM [69], using an estimated sequencing error rate of 0.01. We identified high-confidence genotype calls as those with a conditional probability 10,000× greater than the probabilities of the other possible genotypes. We identified potential somatic SNPs as those with both a high-confidence homozygous and high-confidence heterozygous genotype across the branches.

We notice that the default SNP calling parameters tend to overcall homozygous-reference allele genotypes and that differences in sequencing depth can bias the relative number of heterozygous SNPs detected. To overcome these issues, we re-called genotypes using conditional probabilities using down sampled allele counts. To do this, we first randomly selected a set number of sequencing reads for each library at each potential somatic SNP so that all libraries have the same sequencing depth at all SNPs. Using the down-sampled reads, we calculate the relative conditional probability of each genotypes by dividing the conditional probabilities by the sum of the conditional probabilities of all three potential genotypes. These relative probabilities are then multiplied by the dosage assigned to their respective genotype (0 for RR, 1 for RA, 2 for AA), and the dosage genotype is the sum of these values across all 3 possible genotypes. Discrete genotypes were assigned using the following dosage values: RR = dosage < 0.1; RA = 0.9 < dosage < 1.1; AA = dosage > 1.9. Dosages outside those ranges are assigned a NA discrete genotype. SNPs with an NA discrete genotype or depth below the down sampling level in any branch of a tree were removed from further analysis. We performed three replicates of this procedure for depths of 20, 25, 30, 35, 40, and 45 reads.

PacBio libraries for each branch were sequenced using the PacBio Sequel platform, fastq files aligned to the *P. trichocarpa* var. *Stettler14* reference genome using ngmlr [70], and multi-sample mpileup files generated using in Samtools v1.9 [67] to quantify the allele counts at the potential somatic SNPs. We used a minimum per-sample sequence depth of 20 reads and used an alternate-allele threshold of 0.1 to call a heterozygote genotype in the PacBio data.

To identify high-confidence candidate somatic SNPs, we identified potential somatic SNPs with the same genotypes across branches using both the Illumina-based PacBio-based genotypes, only including SNPs with full data in all branches for both types of genotypes. Of these, we only retained SNPs that are homozygous in a single branch or have a single homozygous-to-heterozygous transition (and no reversion) going from the lowest to highest branches.

### Estimating somatic nucleotide mutation rate

Building on the analytical framework developed in van der Graaf et al. (2015) and Shahryary et al. 2019 (co-submission), we developed *mutSOMA* (https://github.com/jlab-code/mutSOMA), a statistical method for estimating genetic mutation rates in long-lived perennials such as trees. The method treats the tree branching structure as a pedigree of somatic lineages and uses the fact that these cell lineages carry information about the mutational history of each branch. A detailed mathematical description of the method is provided in Additional file 3: Supplementary Text. But briefly, starting from the .vcf* files from $S$ samples representing different branches of the tree, we let $G_{ik}$ be the observed genotype at the $k$th single nucleotide ($k = 1, ..., N$) in the $i$th sample, where $N$ is the effective genome size (i.e., the total number of bases with sufficient coverage). With four possible nucleotides (A, C, T, G), $G_{ik}$ can have 16 possible genotypes in a diploid genome, 4 homozygous (A|A, T|T, C|C, G|G) and 12 heterozygous (A|G, A|T, ..., G|C). Using this coding, we calculate the genetic divergence, $D$, between any two samples $i$ and $j$ as follows:

$$D_{ij} = \sum\nolimits_{k=1}^{N} I(G_{ik}, G_{jk}) N^{-1},$$

where $I(G_{ik}, G_{jk})$ is an indicator function, such that, $I(G_{ik}, G_{jk}) = 1$ if the two samples share no alleles at locus $k$, 0.5 if they share one, and 0 if they share both alleles. We suppose that $D_{ij}$ is related to the developmental divergence time of samples $i$ and $j$ through a somatic mutation model $M_{\Theta}$. The divergence times can be calculated from the coring data (Additional file 2: Table S14). We model the genetic divergence using

$$D_{ij} = c + D_{ij}^{\bullet}(M_{\Theta}) + \epsilon_{ij},$$

where $\epsilon_{ij} \sim N(0, \sigma^2)$ is the normally distributed residual, $c$ is the intercept, and $D_{ij}^{\bullet}(M_{\Theta})$ is the expected divergence as a function of mutation model $M$ with parameter vector $\Theta$. Parameter vector $\Theta$ contains the unknown mutation rate $\delta$ and the unknown proportion $\gamma$ heterozygote loci of the most recent common "founder" cells of samples $i$ and $j$. The theoretical derivation of $D_{ij}^{\bullet}(M_{\Theta})$ and details regarding model estimation can be found in Additional file 3: Supplementary Text. The estimation of the residual variance in the model allows for the fact that part of the observed genetic divergence between any two samples is driven both by genotyping errors as well as by somatic genetic drift as meristematic cells pass through bottlenecks in the generation of the lateral branches.

### Structural variant analysis methods

For structural variant (SV) analysis, PacBio libraries were generated for four branches from the tree 13 and four branches from tree 14 with four sequencing cells sequenced per branch using the PacBio Sequel platform. PacBio fastq files were aligned to the *P. trichocarpa* var. *Stettler* reference genome using ngmlr v.0.2.6 [70] using a value of 0.01 for the "-R" flag. SVs were discovered and called using pbsv (pbsv v2.2.0, https://github.com/PacificBiosciences/pbsv). SV signatures were identified for each sample using "pbsv discover" using the "--tandem-repeats" flag and a tandem repeat BED file generated using trf v4.09 [71] for the *P. trichocarpa* var. *Stettler* genome. SVs were called

jointly for all 8 branches using "pbsv call." The output from joint SV calling changes slightly depending on the order of the samples used for the input in "pbsv call," so four sets of SVs were generated using four different sample orders as input. We used a custom R script to filter the SV output from pbsv [59]. We remove low-complexity insertions or deletions with sequence containing > 80% of a mononucleotide 8-mer, 50% of a single type of binucleotide 8-mer, or 60% of two types of binucleotide 8-mers. We required a minimum distance of 1 kb between SVs. We removed SVs with sequencing coverage of more than three standard deviations above the mean coverage across a sample. After calling genotypes, any SVs with missing genotype data were removed.

Genotypes were called based on the output from pbsv using a custom R script [59]. We required a minimum coverage of 10 reads in all sample and for one sample to have at least 20 reads. We required a minimum penetrance (read ratio) of 0.25 and at least 2 reads containing the minor allele for a heterozygous genotype. We allowed a maximum penetrance of 0.05 for homozygous genotypes. For each genotype, we assigned a quality score based on the binomial distribution-related relative probability of the 3 genotype classes (RR, AR, AA) based on A:R read ratio, using an estimated sequencing error of 0.032, and an estimated minimum allele penetrance of 0.35. For a genotype with a score below 0.9 but with the same genotype at the SV as another sample with a score above 0.98, the score was adjusted by multiplying by 1.67. Any genotypes with adjusted scores below 0.9 were converted to NA. For deletions, duplications, and insertions, 10 representatives in different size classes were randomly selected and the mapping patterns of reads were visually inspected using IGV v2.5.3 [72] to assign scores indicating how well the visual mapping patterns support the SV designation. Scores were defined by the following: "strong," multiple reads align to the same locations in the reference genome that support the SV type and size; "moderate," multiple reads align to the same reference location for one side of the SV but align to different or multiple locations in the region for the other side of the SV; and "weak," reads align to reference locations that indicate a different SV type or much different SV size.

The percent of genic sequence and tandem repeat sequence in deletions and duplications were calculated using the *P. trichocarpa* var. *Stettler* annotation and tandem repeat BED from above, respectively. Genome-wide expectations were derived by separating the genome into 10-kb windows and calculating the percent genic and tandem repeat sequence in each window. The distribution of genic and tandem repeat sequences in deletions and duplications were compared to genome-wide expectations using the Kolmogorov-Smirnov two-sample test (one-sided, $N_{null}$ = 39,151, $N_{del}$ = 10, 433, $N_{dup}$ = 630).

SVs showing variation between branches and identified in all 4 replicates are potential instances of somatic SV mutations or loss-of-heterozygosity gene conversions, and the mapping positions of sequencing reads were visually inspected with IGV [72] to confirm the variation at these SVs.

### MethylC-seq sequencing and analysis

A single MethylC-seq library was created for each branch from leaf tissue. Libraries were prepared according to the protocol described in Urich et al. [73]. Libraries were

sequenced to 150 bp per read at the Georgia Genomics & Bioinformatics Core (GGBC) on a NextSeq500 platform (Illumina). Average sequencing depth was ~ 41.1× among samples (Additional file 2: Table S8).

MethylC-seq reads were processed and aligned using Methylpy v1.3.2 [74]. Default parameters were used except for the following: clonal reads were removed, lambda DNA was used as the unmethylated control, and binomial test was performed for all cytosines with at least three mapped reads. The methylation levels were determined using weighted methylation level, as mC / (mC + uC) where mC and uC are the number of reads supporting a methylated cytosine and unmethylated cytosine, respectively (C/C + T) [44]. The sodium bisulfite conversion rates were benchmarked against spiked in lambda DNA (which was unmethylated). All rates were well over 99% (Additional file 2: Table S8).

### Identification of differentially methylated regions

Identification of differentially methylated regions (DMRs) was performed using Methylpy v1.3.2 [74]. All methylome samples were analyzed together to conduct an undirected identification of DMRs across all samples in the CNN (N = A, C, G, T) context. Default parameters were used. Regions with at least three differentially methylated cytosines (DMS) were combined into raw DMRs. DMS with different directionality (hyper vs hypo) were not combined. Only DMRs that are at least 40 bp long with five or more cytosines (three of which are differentially methylated) with at least one read were used for subsequent analysis. For each DMR, the weighted methylation level was computed as mC / (mC + uC) where mC and uC are the number of reads supporting a methylated cytosine and unmethylated cytosine, respectively [44].

To identify epigenetic variants in these samples, we used a one-sided *z*-test to test for a significant difference in methylation level of DMRs pairwise between branches [59]. For each pair, only DMRs with at least 5% difference in methylation level were used, regardless of underlying context. Resulting *P* values were adjusted using Benjamini-Hochberg correction (*N* = 383,600) with FDR = 0.05 [75], and DMRs are defined by adjusted *P* value ≤ 0.05.

### Identification of methylated regions

For each sample, an unmethylated methylome was generated by setting the number of methylated reads to zero while maintaining the total number of reads. Methylpy DMR identification program [74] was applied to each sample using the original methylome and unmethylated methylome with the same parameters as used for DMR identification. Regions less than 40 bp long, fewer than three DMS, and fewer than five cytosines with at least one read were removed. Remaining regions from all samples were merged using BEDtools v2.27.1 [76].

### Assigning genomic features to DMRs

A genomic feature map was created such that each base pair of the genome was assigned a single feature type (transposable element/repeat, promoter, untranslated region, coding sequence, and intron) based on the previously described annotation. Promoters were defined as 2 kb upstream of the transcription start site of protein-coding

genes. At positions where multiple feature types could be applicable, such as a transposon in an intron or promoter overlapping with adjacent gene, priority was given to transposable elements, untranslated regions, introns, coding sequences, and promoters. Positions without an assignment were considered intergenic. Genomic feature content of each DMR and methylated region was assigned proportionally based on the number of bases in each category.

GO Enrichment analysis of promoter DMRs was run using topGo v2.34.0 with nodeSize = 10 and weighted Fisher's exact for BP, CC, and MF ontologies [77]. Significance was determined for *P* value < 0.001.

### Identification of pseudo-allele methylation

We aimed to categorize the DMRs into three pseudo-allele states: homozygous methylated, heterozygous, and homozygous unmethylated. First, DMRs were filtered on the following criteria: (i) at least 25% change in weighted CG methylation level between the highest and lowest methylation level of the samples; (ii) at least one sample had a CG methylation level of at least 75%; and (iii) at least two "covered" CG positions. A "covered" CG is defined as having at least one read for both symmetrical cytosines in all samples. After filtering, 4488 regions were used for analysis.

For each region in each sample, we next categorize the aligned reads overlapping the region [59]. If at least 35% of its "covered" CG sites are methylated, the read is categorized as methylated. Otherwise, it is an unmethylated read. Finally, we define the pseudo-allele state by the portion of methylated reads; homozygous unmethylated: ≤ 25%, heterozygous: > 25% and < 75%, and homozygous methylated: ≥ 75%.

The null distribution was created by randomly shuffling the filtered DMRs in the genome such that each simulated region is the same length as the original and it has at least two "covered" CGs. The above procedure was applied and number of epigenotype changes was determined. This was repeated for a total of 10 times.

The following special classes of DMRs were identified: highly variable, single loss, single gain, and tree specific. A DMR is highly variable if there were pseudo-allele changes between all adjacent branches. A DMR is single loss if all but one branch was homozygous methylated, and one was homozygous unmethylated. Similarly, a DMR is single gain if all but one branch was homozygous unmethylated and one branch was homozygous methylated. Finally, a DMR is "tree specific" if all tree 13 branches were homozygous unmethylated and all tree 14 branches were homozygous methylated or vice versa.

### Estimating somatic epimutation rate

We previously developed a method for estimating "germline" epimutation rates in *A. thaliana* based on multi-generational methylation data from mutation accumulation lines [34]. In a companion method paper to the present study (Shahryary et al. 2019, co-submission), we have extended this approach to estimating somatic epimutation rates in long-lived perennials such as trees using leaf methylomes and coring data as input. This new inference method, which we call *AlphaBeta*, treats the tree branching structure as a pedigree of somatic lineages using the fact that these cell lineages carry information about the epimutational history of each branch. *AlphaBeta* is implemented

as a bioconductor R package (http://bioconductor.org/packages/devel/bioc/html/Alpha-Beta.html). The results detailing the significance of epimutation accumulation as described in Shahryary et al. (2019) are contained in the Additional file 2: Table S9. Using this approach, we estimate somatic epimutation rates for individual CG, CHG, and CHH sites independently, but also for regions. For the region-level analysis, we first use the differentially methylated regions (DMRs) identified above. Sampling from the distribution of DMR sizes, we then split the remainder of the genome into regions, which we refer to as "non-DMRs." Per sample, we aggregate the total number of methylated Cs and unmethylated Cs in each region corresponding to a DMR or a non-DMR and used these counts as input for *AlphaBeta*.

In a replication experiment for tree 13 and tree 14, we sequenced the methylomes of leaves from seven branches sampled from the same branches (3 samples from tree 14 and 4 samples from tree 13). Initial quality control of the methylome data revealed that five of the seven samples (14–3.1, 13–1.1, 13–2.1, 13–3.1, 13–5.1) clustered well with their branch-matched replicates. However, two of the samples from tree 14 (14–4.1 and 14–5.1) revealed contamination making them unusable. Therefore, they were excluded from further analysis. Using the remaining five replicates, we re-estimated the genome-wide gain and loss rates and found that they were very similar to those obtained with the original samples (Additional file 1: Fig. S6a-d). In addition to a "complete" tree analysis (involving samples from both tree 13 and tree 14), we also examined epimutation accumulation in tree 13 alone (Additional file 1: Fig. S6e). Similar to the trends we observed with the original samples (Additional file 1: Fig. S6e-h), 5mC divergence increased as a function of age in the replicate data, although these accumulation patterns are not significant due to the small sample sizes ($N = 4$).

### mRNA-seq sequencing and analysis

Total RNA was extracted from leaf tissue in each branch using the Direct-zol RNA MiniPrep Plus kit (Zymo Research) with Invitrogen's Plant RNA Reagent. Total RNA quality and quantity were assessed before library construction. Strand-specific RNA-seq libraries were constructed using the TruSeq Stranded mRNA LT kit (Illumina) following the manufacturer's instructions. For each sample, three independent libraries (technical replicates) were constructed. Libraries were sequenced to paired-end 75-bp reads at the GGBC on a NextSeq500 platform (Illumina). Summary statistics are included in the Additional file 2: Table S10.

For analysis, first, paired-end reads were trimmed using Trimmomatic v0.36 [78]. Trimming included removing TruSeq3 adapters, bases with quality score less than 10, and any reads less than 50 bp long. Second, remaining reads were mapped to the *Stettler* genome with HiSAT2 [79] using default parameters except to report alignments for transcript assemblers (--dta). The HiSAT2 transcriptome index was created using extracted splice sites and exons from the gene annotation as recommended. Last, transcriptional abundances for genes in the reference annotation were computed for each sample using StringTie v1.3.4d [80]. Default parameters were used except to limit estimates to reference transcripts. TPM (transcripts per million) values were outputted to represent transcriptional abundance.

### Identification of differentially expressed genes

Differentially expressed genes (DEGs) were identified using DeSeq2 v1.22.2 [49]. The count matrix was extracted from StringTie output files and the analysis was performed using the protocol (ccb.jhu.edu/software/stringtie/index.shtml?t = manual#deseq). Abundances for all samples were joined into one DESeq dataset with $\alpha = 0.01$. Gene abundance was compared between all samples pairwise. In each pair, a gene was considered differentially expressed if the adjusted $P$ value $\leq 0.01$ and the $\log_2$-fold change $\geq 1$. Genes differentially expressed in any pair were included for subsequent analysis.

### Overlap of DMRs and DEGs

We identified DMRs which overlapped the promoter region (2 kb upstream of transcription start site) and gene body of annotated genes. For each DMR-gene pair, we computed Pearson's product moment correlation coefficient between the weighted methylation level of the DMR and average gene abundance among replicates in TPM. Next, looking only at genes which were previously identified as differently expressed, we performed two-sided Pearson's correlation test for each DMR-DEG pair to test for statistically significant correlations. Resulting $P$ values were multiple test corrected with Benjamini-Hochberg correction ($N = 382$, FDR = 0.05) [75]. Adjusted $P$ values $\leq 0.05$ were considered significantly correlated.

### Supplementary information

**Additional file 1: Fig. S1.** Photographs of the trees used in this study. Photographs of tree 4 (a), tree 9 (b), tree 13 and 14 (c), and tree 15 (d) with branches labeled. Leaf samples were collected from each branch. **Fig. S2.** Schematic drawings of additional trees in the study. Schematic drawings of tree 4 (a), tree 9 (b), tree 13 and 14 (c), and tree 15 (d) with estimated terminal branch ages and age where branch meets the main stem (gray italic). Leaf samples were collected from each branch for genomic sequencing libraries. **Fig. S3.** Duplications contain a higher proportion of genic sequences and deletions contain a higher proportion of repeat sequence. a) For deletions (Del, green) and duplications (Dup, purple) structural variants grouped by size, distribution of the proportion of the SV sequence that overlaps with an annotate gene. Same as a except proportion of the SV sequence that overlaps transposons and repeat sequences. Genome-null (gray) is measured for 10-kb windows across the genome. Diamond represents the group mean. Number of SVs in each group is specified above b. **Fig. S4.** Genome weighted methylation levels. Genome-wide weighted methylation level for mCG (red), mCHG (blue), and mCHH (yellow) for samples in tree 13 and tree 14. **Fig. S5.** Somatic epimutation rates for single sites, regions, and by genomic feature in the CHH context. Methylation divergence by branch time divergence for single sites and regions (a) and genomic features (b). c) Estimated methylation gain rate, α, by feature. d) Estimated methylation loss rate, β, by feature. e) Estimated ratio of loss to gain, β/α. An F-test was used comparing the neutral model vs null model (Supplementary Text). See Table S9 for $P$ values. Error bars represent bootstrapped 95% confidence intervals of the estimates. If there is no significant effect of branch age for the feature, it is marked n.s. Abbreviations: Pro, promoter (2 kb upstream of TSS); TE, transposable elements and repeats; and IGR, intergenic regions. **Fig. S6.** Comparison of original and replicate methylome data sets. (a) mCG divergence of the original data vs the replicated data set from the same tree. F- and $P$ value show significant accumulation of mCG changes over time in both data sets. (b) estimated rate of methylation gain and (c) loss; (d) ratio of loss over gain of methylation. (e) mCG divergence, (f) gain rate, (g) loss rate and (h) loss over gain ratio for only branch 13 of both the original and the replicate data set. The F- and $P$ values in (e) suggest no significant time-dependent accumulation of epimutations among leafs of only branch 13. Error bars in (b), (c), (f), (g) represent the standard errors generated during bootstrapping. **Fig. S7.** Pseudo allele states of DMRs among samples. a) Pseudo allele state of each tested DMR ($N = 4488$) for each branch. b) Branches 13.1 and 14.2 proportionally have more homozygous methylated pseudo alleles than the older branches. Possible pseudo allele states are homozygous methylated (dark green), heterozygous (medium green), and homozygous unmethylated (light green). **Fig. S8.** Gene expression of differentially expressed genes is rarely correlated to methylation level of nearby differentially methylated regions. Each point represents a differentially expressed gene-differentially methylated region pair where the DMR is in the gene body or within 2-kb upstream. Correlation is the Pearson's correlation between gene expression, average of replicates as TPM, and weighted methylation level. Pearson's correlation test, two-sided, was performed on each pair then multiple test corrected using Benjamini-Hochberg ($N = 382$, FDR = 0.05). Red dashed line is the significance threshold, adjusted $P$ value $\leq 0.05$. Significant DEG-DMR pairs are colored red. **Fig. S9.** Syntenic *Nisqually* marker placements on the *Populus trichocarpa* var. *Stettler* chromosomes. Each point represents a *Nisqually* marker positioned along the *Nisqually* chromosome along the x-axis and *Stettler* chromosome along the y-axis.

**Additional file 2: Table S1.** High-confidence SNPS identified in Tree 13 with corresponding branch genotypes. In the genotypes, "RR" is homozygous for reference allele and "RA" is heterozygous for alternative allele. **Table S2.** High-confidence SNPS identified in Tree 14 with corresponding branch genotypes. In the genotypes, "RR" is homozygous for reference allele and "RA" is heterozygous for alternative allele. **Table S3a.** Fold-Enrichment of SNPs in different genomic feature types. **Table S3b.** Fold-Enrichment of SNPs in different transposable element classes. **Table S4.** Nucleotide mutation rate estimates for five filtering depths and multiple replicates. GS is the effective genome size. **Table S5.** Total PacBio sequencing output for the branches used for structural variation analysis. Table S6. Counts of SVs separated by type and size. Count is the mean of four replicates of 'pbsv call' with standard deviation in parentheses. **Table S7.** Support for SV designation of random subset of SVs based on visual evaluation of read mapping patterns in IGV. Percentages, by row, in parentheses. **Table S8.** Whole-genome bisulfite sequencing summary statistics. **Table S9.** Calculated epimutation rates by sequence context and genomic feature. Alpha is the rate of gaining methylation. Beta is the rate of losing methylation. F-test compares the neutral model (degrees of freedom 23) vs null model (d.o.f. 27). **Table S10.** mRNA-seq library and mapping statistics. **Table S11.** Genomic libraries included in the *Populus trichocarpa* var. Stettler14 genome assembly and their respective assembled sequence coverage levels in the final release. *Average read length of PacBio reads. **Table S12.** PacBio library statistics for total yield of the 64 chips included in the *Populus trichocarpa* var. Stettler14 genome assembly and their respective assembled sequence coverage levels. **Table S13.** Summary statistics of the raw output of the MECAT whole genome shotgun assembly. The table shows total contigs and total assembled base pairs for each set of scaffolds greater than the size listed in the left-most column. **Table S14.** Whole-genome bisulfite sequencing summary statistics for replicate samples.

**Additional file 3.** Supplementary text. Includes an expanded description of how epimutation rates were estimated.

**Additional file 4.** Review history.

## Acknowledgements

We thank the JGI and collaborators for pre-publication access to *P. trichocarpa* v4.0 genome sequence for chromosome scale ordering of the *Stettler* genome and use of the RNA-seq data from the JGI Plant Gene Atlas for annotation. Sample collection was supported by the Center for Bioenergy Innovation (CBI). CBI is Bioenergy Research Centers supported by the Office of Biological and Environmental Research in the US Department of Energy Office of Science. We also thank Dr. Pankaj Jaiswal for use of additional RNA-seq data included in the annotation.

## Review history

The review history is available as Additional file 4.

## Peer review information

## Authors' contributions

RJS, FJ, GAT, RS, and JS conceived and designed the experiments. JG, SS, KB, KL, CA, AL, DK, JT, and RW performed the data generation. BTH, JD, MCT, YS, RH, SM, JJ, PPG, and FJ performed the data analysis. BTH prepared the figures and manuscript. BTH, DWH, GAT, FJ, and RJS wrote and revised the manuscript with input from all authors. All authors read and approved the final manuscript.

## Funding

This study was supported by the National Science Foundation (IOS-1546867) to RJS and JS and the National Institutes of Health (R01-GM134682) to RJS and DWH. FJ and RJS acknowledge support from the Technical University of Munich-Institute for Advanced Study funded by the German Excellent Initiative and the European Seventh Framework Programme under grant agreement no. 291763. FJ is also supported by the SFB/Sonderforschungsbereich924 of the Deutsche Forschungsgemeinschaft (DFG). RJS is a Pew Scholar in the Biomedical Sciences, supported by The Pew Charitable Trusts. BTH was supported by the National Institute of General Medical Sciences of the National Institutes of Health (T32GM007103). The work conducted by the U.S. Department of Energy Joint Genome Institute is supported by the Office of Science of the U.S. Department of Energy under Contract No. DE-AC02-05CH11231. Sequencing in this project was partially supported by a JGI community sequencing project grant CSP1678 to RS and GAT.

## Availability of data and materials

Raw sequence data used for genome assembly, resequencing, and identification of structural variation of individual branches are available at NCBI SRA under accession PRJNA516415 [81]. Raw sequence data for whole-genome bisulfite sequencing and mRNA-sequencing are available at GEO under accession GSE132939 [82].
Custom analysis scripts used in this study are available at GitHub (https://github.com/schmitzlab/somatic-epigenetic-mutation-poplar) and at Zenodo with the identifier doi:https://doi.org/10.5281/zenodo.4000597 [59].

## Ethics approval and consent to participate

Not applicable.

## Consent for publication

Not applicable.

## Competing interests

The authors declare that they have no competing interests.

## Author details

[1]Institute of Bioinformatics, University of Georgia, Athens, GA, USA. [2]Helmholtz Center Munich, German Research Center for Environmental Health, Institute of Computational Biology, Neuherberg, Germany. [3]European Research Institute for the Biology of Ageing, University of Groningen, University Medical Centre Groningen, Groningen, The Netherlands. [4]TUM School of Life Sciences Weihenstephan, Technical University of Munich, Freising, Germany. [5]Department of Plant Sciences, Technical University of Munich, Liesel-Beckmann-Str. 2, Freising, Germany. [6]Institute for Advanced Study (IAS), Technical University of Munich, Lichtenbergstr. 2a, Garching, Germany. [7]HudsonAlpha Institute of Biotechnology, Huntsville, AL, USA. [8]Department of Energy Joint Genome Institute, Walnut Creek, CA, USA. [9]Department of Molecular Genetics, Weizmann Institute of Science, Rehovot, Israel. [10]Arizona Genomics Institute, School of Plant Sciences, University of Arizona, Tucson, AZ, USA. [11]Department of Genetics, University of Georgia, Athens, GA, USA. [12]The Center for Bioenergy Innovation, Oak Ridge National Laboratory, Oak Ridge, TN, USA.

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

## 

