## [**Additional file 4.** Review history. · Genome Biology]

Review History

First round of review

Reviewer 1

Are you able to assess all statistics in the manuscript, including the appropriateness of statistical tests used? Yes, and I have assessed the statistics in my report.

Comments to author:

I have carefully gone through these two manuscripts "The somatic genetic and epigenetic mutation rate in a wild long-lived perennial *Populus trichocarpa*" and "AlphaBeta: Computational inference of epimutation rates and spectra from high-throughput DNA methylation data in plants" (referred to as Paper 1 [Hofmeister et al.] and Paper 2 [Shahryary et al.] hereafter). Due to the close relationship between these two, I would like to give my comments as a whole for the two papers.

Both papers focus on epigenetic variation in the ancestors and their progenies or in the age-different branches from the same tree. The epimutation variation (roughly as DNA methylation used in these papers) has been widely studied among different stages of development in a plant, various organs of a plant, and the individual plants under different environments. Similarly, these two studies proposed the pedigree-based or age-based variation of epimutation. Generally speaking, the epimutation variation are influenced by many factors so little studies have revealed clear patterns with biological significance. If such clear patterns could be obtained between pedigree (or age) and epimutation, those studies could be very important scientifically.

These two studies have made positive progress on such patterns. For example, they provide evidences to show neutral pattern of epimutation accumulation in context CG in generation by generation in a pedigree and age-increased branches. Because this pattern is present in all analyzed species and in both pedigree- and age-related samples, they conclude this pattern is deeply conserved in plant, which can be used as a molecular clock for age-dating trees. The significant accumulation of epimutation (basically the 5mC methylation) as time increase provides insight to understand epimutation variation. However, the biological significance is not so clear and such patterns revealed by these studies are not so strong.

Comments on both papers:

1) Consequences of the neutral pattern of epimutation accumulation. The most striking finding claimed by authors is the neutral pattern which is conserved in plant species. (e.g., in Discussion: "One of most striking findings was the close similarity in the epimutation landscapes between these very different systems" in page 21). Based on this finding, epimutation is expected to accumulate much more when a branch was separated much earlier from the others. In fact, all new leave in the branches of the tree 13 should be equal separated to all the samples from tree 14. Therefore, the comparison between the two trees is an extreme contrast which should result in much more accumulation of epimutations. Similarly, when the epimutation accumulates as generation increases, a logical inference is that the epimutation accumulation should be much more between two close-related (which relationship can be easily identified by a phylogenetic tree) individual plants in a population (e.g., different *Arabidopsis* individuals from the same accession), because the two plants obviously came from a common ancestor plant with hundreds or thousands of generations. This comparison should be very easy to do and should produce much more strong patterns.

These kinds of comparisons between extreme-samples should provide much more robust results if the pattern revealed by these papers are real.

2) These two manuscripts overlap to some extent. The Paper 1 develop a new bioinformatics tool and use poplar data to test for its robustness. However, many of the results about mutation and epimutation coincide with previous reports and may not be unexpected. So, I suggest the authors are better to consider combining these two manuscripts into one.

3) Biological significance of epimutation accumulation. Based the results in these studies, epimutation accumulation will lead to DNA methylation diversity. If it is true, the DNA methylation diversity (or epimutation) is somewhat equivalent to DNA diversity (or mutation) which will also accumulate as generation by generation or by increased age. Two kinds of diversity (mutations) could have some common features. Is it possible to discuss a few sentences to give some possible biological significance?

Comments Specifically on Paper1 [Hofmeister et al.]

The authors carried extensive analyses on somatic changes of the long-lived perennial poplar. Their results tell an intriguing fact that there seems to be little difference between the short-lived Arabidopsis and the long-lived poplar on the rates of both the genomic level and the epi-genomic level during the mitotic cell divisions. The overall experiments and methodologies are generally well documented, with a part given in the co-submitted paper. The results are also well presented and the paper is easy to follow.

Some issues which may need further clarifications:

1) Do we have any external validation or cross-validation on all the mutations/epimutations identified?

2) Page 9 Line 197. Does the 40M genome space means only 10% of the genome is analyzed in mutation calling? Could this imply a very high false negative rate due to too stringent filtering? If indeed only this part of genome could be reliable for mutation calling, is this part representative for the whole genome?

3) Page 11 Line 251. About the declaration of the age-dependent of cumulative spontaneous methylation changes, do we have any statistical supports? I see lines in Fig. 3a-d but did not find details of how these lines were obtained. Are those regression lines? If so what's the slope as well as the interception. I assume these are similar plots as Fig. S5a and S5b. But for dots at two ends we can always obtain a line... The lines in Fig. S5 look rather horizontal to me, i.e., neither increasing nor decreasing. I suggest more details are required to interpret these plots, and proper statistical tests are required to prove the age-dependent effect (e.g., the slopes are truly positive).

4) Page 14 Line 315. "... a common feature of epimutations in plants" needs citations.

5) Page 15 Line 339. This paragraph is rather descriptive with no supporting data shown in the paper... Are those methylated cytosines clustered together and the investigated C-DMRs (4,488 regions used here?) happen to be the same clustered regions? If true, the claim of same mechanism would become meaningless as we may just observe the same thing for both region-level and base-level.

- 6) Page 16 Line 356. As commented above for Fig. 3a-d, Fig. 5a is an example that a line with positive slope with no significant associations (as shown by the authors) ... Please also clearly state the statistics used here in the figure legend.
- 7) Page 16 Line 358. Since this claim could be confounded by age-estimation as well as different regions assessed for mutation/epimutation/transcriptional changes. Why not directly test the correlation between three changes? This could give more valuable indications of whether two or three changes are "coupled" with each other. The following correlation between DMRs and DEGs is a good proof for this.
- 8) Page 16 Line 365-367. Are those correlations significant? Please provide these details here (I see p-values for Fig. 5c and 5d, but none for 5b).
- 9) Page 17 Line 378. The claim of independence between methylation and transcriptional changes are weak to me based on the analysis here. The comparisons are either indirect or with few statistical supports.
- 10) Page 17 Line 379. I don't see very clear how this independent correlation was connected to the neutrality of somatic epimutations. Do we assume the transcriptional changes are under strong selections?
- 11) Page 20 Line 441. Are those young leaves or old leaves? Are all leaf samples collected at the same time point?
- 12) Page 22 Line 498. How were the scaffolds oriented, ordered, and joined? Using the syntenic sequences from Nisqually assembly?
- 13) Page 22 Line 501. "Althap" = Alternative haplotype?
- 14) Page 25 Line 551. "... were removed and weak gene models" needs rephrase.
- 15) Page 26 Line 591. "... fastq files were aligned ...". Line 592 "... mileup files were generated ..."
- 16) Page 27 Line 597. I am a bit confused about how the mutation was actually defined or called, as I see several sites have only "RR" genotypes across all branches (Table S1-2). Shouldn't we expect at least one branch has a different genotype?
- 17) Page 27 Line 616. It seems mutSOMA models all possibilities of genotype changes, i.e., "homozygous to homozygous, homozygous to heterozygous, heterozygous to homozygous and heterozygous to heterozygous genotypes". But only "homozygous to heterozygous" were considered when calling mutations. Does this influence the rate estimation?
- 18) Page 33 Line 746. Any reason for the priority or just arbitrary?

Reviewer 2

Are you able to assess all statistics in the manuscript, including the appropriateness of statistical tests used? Yes, and I have assessed the statistics in my report.

Comments to author:

The authors analyzed genomes, DNA methylation and transcription of samples of different ages collected from 2 stems of ONE 330 years-old *Populus trichocarpa* tree. Somatic mutations and epimutations are analyzed with the transcriptional variations. The authors present an interesting work and may have some good results to share with the community. My major concerns are the small sample size (Tree=1) and the lack to replicates. The other include that the Methods are not very clear, mixing sequence contexts for DMR analysis. The manuscript reports several individual cases that may not always reveal global pattern correctly.

Major

Sample size is only one tree and there is no replicate for each branch. Although the result is interesting I would still ask for more trees and replicates if it is possible in any way to show reproducibility and the consistency.

What is the fold enrichment of SNPs in TE and in promoters? For those found in TEs, are there any preference of TE families (young/old, long/short)?

Are the distribution of the SNPs the same between the 2 stems? e.g., genome wide distributions, locations, genes covering these SNPs etc?

The % mapped bisulfite reads seems a bit low (Table S7), considering the customized reference genome is used, and the high mRNA-seq mapping rates. Any possible reasons offered?

How the methylation levels are estimated should be clearly described, as well as how the bisulfite unconversion rate was estimated from lambda.

Pg 12, if mCG and mCHG are cumulative across somatic development and point to a shared meristematic origin, what about mCHH? Is mCHH epimutation not accumulative at all? Also, on Fig 3C it seem there are few lines (regions) showing flat (non-accumulative), what are they and why?

P12, L168, what are the estimates reported in *A. thaliana* MA lines?

P13 L282, does the CMT expression vary with the gaining or losing methylation?

P13 L287, so the statement that "epimutations are not a result of biased reinforcement of DNA methylation during sexual reproduction or environment/genetic variation, but instead a feature of DNA methylation maintenance through mitotic cell divisions." sounds like a model rather than a solid conclusions? If so, the statement in Abstract should be tuned down. If it is more than a working model then perhaps more direct evidences should be expected/presented.

P14 L298, I am not sure why the DMR are detected regardless the sequence contexts. Clearly CG, CHG and CHH have very different methylation levels ranging from 36% in mCG to 2% in mCHH, and their behaviors in this study are apparently different. I would analyze the DMR by the sequence context rather than combining all.

Not very clear on Fig 4a how to observe age-dependent accumulation of DMR

Are these DMR distributed randomly or there are hot spots? Since some of them prefer promoters, any specific genes are they targeting?

In DMR methods it say all samples are analyzed together to obtain DMR. If so, these are methylation variable regions among these samples. How does that compare to sets of DMR from pairwise comparisons between samples?

Also, are the direction of DMS (hyper or hypo) considered within one DMR? Do they have to be concordant? -- The Methods should cover all steps in details.

Perhaps more global analyses should be carried out for the allelic methylation changes in addition to reporting individual cases in Figure 4defg. How strong or prevalent are the allelic methylation changes?

How do we see from Fig 5a that the accumulation of genetic and epigenetic changes are largely uncoupled? Both show accumulation, no?

Figure 5b, would it make more sense again to separate the CG, CHG, and CHG DMR here to study DMR-DEG correlation, as both the direction and the contexts matter?

Minor

The color key is missing on Fig 3abcd. I can't tell which line is which.

Authors response

Reviewer 1

Comments on both papers:

1) Consequences of the neutral pattern of epimutation accumulation. The most striking finding claimed by authors is the neutral pattern which is conserved in plant species. (e.g., in Discussion: "One of most striking findings was the close similarity in the epimutation landscapes between these very different systems" in page 21). Based on this finding, epimutation is expected to accumulate much more when a branch was separated much earlier from the others. In fact, all new leave in the branches of the tree 13 should be equal separated to all the samples from tree 14.

To be clear, divergence times between leaves from tree 13 and tree 14 depend on the specific branches from which they were sampled on each of the trees. Divergence time is calculated by tracing back time along the branches from which they were sampled and down along the main stems. Since the branch points of the branches have different ages (as do the branches themselves) the divergence times between branches is not equal.

Therefore, the comparison between the two trees is an extreme contrast which should result in much more accumulation of epimutations.

That is true. We do want to note that we find similar epimutation rates and accumulation patterns by looking only at within-tree (rather between tree) comparisons.

Similarly, when the epimutation accumulates as generation increases, a logical inference is that the epimutation accumulation should be much more between two close-related (which relationship can be easily identified by a phylogenetic tree) individual plants in a population (e.g., different *Arabidopsis* individuals from the same accession), because the two plants

obviously came from a common ancestor plant with hundreds or thousands of generations. This comparison should be very easy to do and should produce much more strong patterns. These kinds of comparisons between extreme-samples should provide much more robust results if the pattern revealed by these papers are real.

A problem with inferring epimutation rates from phylogenetic trees in natural populations is that 5mC divergence between *A. thaliana* accessions is also driven by genetic polymorphisms, which makes it very challenging (if not impossible) to relate 5mC diversity to epimutation rates. Recent approaches based on the analysis of the methylation site frequency spectrum (mSFS) provide a step in this direction. But they only estimate the population epimutation rate (which is the product of the per site per haploid genome epimutation rates and the effective population size). See Vidalis et al. 2016, Genome Biology.

2) These two manuscripts overlap to some extent. The Paper 1 develop a new bioinformatics tool and use poplar data to test for its robustness. However, many of the results about mutation and epimutation coincide with previous reports and may not be unexpected. So, I suggest the authors are better to consider combining these two manuscripts into one.

Upon consultation with the editor, we have been asked to resubmit both manuscripts as originally submitted.

3) Biological significance of epimutation accumulation. Based the results in these studies, epimutation accumulation will lead to DNA methylation diversity. If it is true, the DNA methylation diversity (or epimutation) is somewhat equivalent to DNA diversity (or mutation) which will also accumulate as generation by generation or by increased age. Two kinds of diversity (mutations) could have some common features. Is it possible to discuss a few sentences to give some possible biological significance?

This is a great point. We have expanded the discussion as follows to discuss the possible biological significance of epimutation accumulation.

Comments Specifically on Paper1 [Hofmeister et al.]

The authors carried extensive analyses on somatic changes of the long-lived perennial poplar. Their results tell an intriguing fact that there seems to be little difference between the short-lived Arabidopsis and the long-lived poplar on the rates of both the genomic level and the epi-genomic level during the mitotic cell divisions. The overall experiments and methodologies are generally well documented, with a part given in the co-submitted paper. The results are also well presented and the paper is easy to follow.

We appreciate your time evaluating our manuscript and providing useful feedback to make it better.

Some issues which may need further clarifications:

1) Do we have any external validation or cross-validation on all the mutations/epimutations identified?

To validate the epimutations we have added replicates for tree 13 and 14. These results validate our original conclusions.

The identified mutations were validated by comparing results from PacBio data and Illumina data, as described in “SNP calling methods” section.

2) Page 9 Line 197. Does the 40M genome space means only 10% of the genome is analyzed in mutation calling? Could this imply a very high false negative rate due to too stringent filtering? If indeed only this part of genome could be reliable for mutation calling, is this part representative for the whole genome?

This is a correct evaluation of the method. The percent of the genome assessed will not affect the overall rate estimated, as we find that the regions used in the final analysis are representative broadly of diverse sequences in the genome assembly.

3) Page 11 Line 251. About the declaration of the age-dependent of cumulative spontaneous methylation changes, do we have any statistical supports? I see lines in Fig. 3a-d but did not find details of how these lines were obtained. Are those regression lines? If so what's the slope as well as the interception. I assume these are similar plots as Fig. S5a and S5b. But for dots at two ends we can always obtain a line... The lines in Fig. S5 look rather horizontal to me, i.e., neither increasing nor decreasing. I suggest more details are required to interpret these plots, and proper statistical tests are required to prove the age-dependent effect (e.g., the slopes are truly positive).

Results of statistical test to prove significance were included in the original submission in supplementary table S8. To make this clearer we have added the following statement to the manuscript, “The results detailing the significance of epimutation accumulation as described in Shahryary et al. 2019 are contained in the supplementary Table S9.”

We have also added a better description to the figure legend. “The dots show the individual observed divergences, whereas the line represents the fit of the data to the model.”

4) Page 14 Line 315. "... a common feature of epimutations in plants" needs citations. Thank

you for noting this. We have added citations to support this statement.

5) Page 15 Line 339. This paragraph is rather descriptive with no supporting data shown in the paper... Are those methylated cytosines clustered together and the investigated C-DMRs (4,488 regions used here?) happen to be the same clustered regions? If true, the claim of same mechanism would become meaningless as we may just observe the same thing for both region-level and base-level.

We apologize for not referencing the data in the original version. We have now added the reference to table S9 which shows the rates of regions and figure 3 where the region-level rates are seen alongside the other epimutation rate estimates. To show that the cytosines in C-DMRs are not driving the epimutation rates for single CG and CHG sites, we have included that only 17% of CGs as well as 17% of CHGs are located inside the identified C-DMRs. These results show that the regional CG sites are not driving the observed rates, as CG sites outside of these regions also accumulate changes over time. The reviewer is correct that this paragraph focuses on regional differences in DNA methylation, as opposed to single cytosines. The reason for this analysis is that distinct DNA methylation pathways are responsible for maintaining DNA methylation at different regions of the genome. For example, gene body methylation is primarily maintained by activity of MET1, whereas regions with CG and non-CG methylation are maintained by MET1, CMT3 and DRM2. The goal of this paragraph was to determine if regional

differences varied in epimutation spectrum and frequency. However, the reviewer is correct that we presented a result without referring to the data. We have referenced the supplementary table with the data and we clarified the paragraph by explaining the goal and conclusions better.

6) Page 16 Line 356. As commented above for Fig. 3a-d, Fig. 5a is an example that a line with positive slope with no significant associations (as shown by the authors) ... Please also clearly state the statistics used here in the figure legend.

We apologize for not being clearer. The statistical test used is in table S8 and was included in our original submission. To improve clarity, we have added the statistical test used to the figure legend and referred to table S9 for the p-value.

7) Page 16 Line 358. Since this claim could be confounded by age-estimation as well as different regions assessed for mutation/epimutation/transcriptional changes. Why not directly test the correlation between three changes? This could give more valuable indications of whether two or three changes are "coupled" with each other. The following correlation between DMRs and DEGs is a good proof for this.

See response to point 10.

8) Page 16 Line 365-367. Are those correlations significant? Please provide these details here (I see p-values for Fig. 5c and 5d, but none for 5b).

These correlations are not significant. As indicated in line 370 of the original submission, only 4 were significant.

9) Page 17 Line 378. The claim of independence between methylation and transcriptional changes are weak to me based on the analysis here. The comparisons are either indirect or with few statistical supports.

See response to point 10.

10) Page 17 Line 379. I don't see very clear how this independent correlation was connected to the neutrality of somatic epimutations. Do we assume the transcriptional changes are under strong selections?

We apologize for not being clearer. We have rephrased this section to the following: "Our results show that divergence in DNA methylation is not accompanied by transcriptional divergence during tree aging, probably as a result of gene expression being much more dynamic and responsive to current environments."

To address the questions in point 7, 9 and 10 we have added the following:

We have fitted the transcriptional changes to our model of time-dependent accumulations and have found it to be insignificant. The p-value and F-statistic for this test were included in Fig. 5a. Additionally, we have conducted a linear regression analysis between transcription and methylation changes over time, which has not shown a statistically significant coupling (p-value = 0.1293). This information was added to the main text as follows: "Linear regression analysis has shown no statistically significant relationship between transcription and methylation over time (p-value = 0.1293)."

11) Page 20 Line 441. Are those young leaves or old leaves? Are all leaf samples collected at the same time point?

Young emerging leaves were harvested within a few hours from each tree. We have added a better description of this information to the methods section.

“Simultaneously with stem coring, young leaf samples were collected from the tips of each of the branches from the selected five trees within a few hours from each tree”.

12) Page 22 Line 498. How were the scaffolds oriented, ordered, and joined? Using the syntenic sequences from Nisqually assembly?

We have amended the genome assembly methods section to reflect these additional details as follows:

“In the syntenic marker FASTA file, each record identifier carried information pertaining to the Nisqually chromosome where the sequence was extracted, as well as the position in the chromosome. These markers, along with the annotated primary transcripts from Nisqually were aligned to the Poplar var 14.5 assembly using BLAT. The chromosome/position information was used to identify misjoins in the assembly. Once the misjoins were corrected, the scaffolds were ordered and oriented using the positional information contained in the syntenic markers/genes.”

13) Page 22 Line 501. "Althap" = Alternative haplotype?

This is correct. We have clarified this in the text.

14) Page 25 Line 551. "... were removed and weak gene models" needs rephrase.

Corrected

15) Page 26 Line 591. "... fastq files were aligned ...". Line 592 "... mileup files were generated ..."

We have corrected this typo from “mileup” to “mpileup”.

16) Page 27 Line 597. I am a bit confused about how the mutation was actually defined or called, as I see several sites have only "RR" genotypes across all branches (Table S1-2). Shouldn't we expect at least one branch has a different genotype?

Thank you so much for catching this mistake. We uploaded the incorrect SNP lists. The correct lists have now replaced original Table S1-2.

17) Page 27 Line 616. It seems mutSOMA models all possibilities of genotype changes, i.e, "homozygous to homozygous, homozygous to heterozygous, heterozygous to homozygous and heterozygous to heterozygous genotypes". But only "homozygous to heterozygous" were considered when calling mutations. Does this influence the rate estimation?

Indeed, we are conditioning our analysis on loci that are most likely in a homozygous state in the genome of the ‘founder’ cells, and infer the mutation rates from homozygote to heterozygote transitions. Since the rates are normalized by the ‘effective genome size’ this should not affect the rate estimates.

18) Page 33 Line 746. Any reason for the priority or just arbitrary?

We made a mistake in the original submission; transposons and repeats were given the highest priority. We chose this ordering based on percentage of genome and potential for overlap.

Reviewer #2: The authors analyzed genomes, DNA methylation and transcription of samples of different ages collected from 2 stems of ONE 330 years-old *Populus trichocarpa* tree. Somatic mutations and epimutations are analyzed with the transcriptional variations. The authors present an interesting work and may have some good results to share with the community. My major concerns are the small sample size (Tree=1) and the lack to replicates. The other include that the Methods are not very clear, mixing sequence contexts for DMR analysis. The manuscript reports several individual cases that may not always reveal global pattern correctly.

Major

Sample size is only one tree and there is no replicate for each branch. Although the result is interesting I would still ask for more trees and replicates if it is possible in any way to show reproducibility and the consistency.

We originally planned to sequence two trees, but to our surprise we learned they were the same tree. To address this comment from the reviewer we have added replicates for all branches of tree 13 and 14 that had sample available. Reanalysis of these data confirm the validity of our original conclusions.

What is the fold enrichment of SNPs in TE and in promoters? For those found in TEs, are there any preference of TE families (young/old, long/short)?

We have added additional detail about the SNP enrichment in the manuscript text (lines 215217) and in a supplementary table (Table S3).

Are the distribution of the SNPs the same between the 2 stems? e.g., genome wide distributions, locations, genes covering these SNPs etc?

The distribution of SNPs is different between the two stems. Tree 14 has more SNPs in transposable elements and fewer in genes compared to Tree 13 and genome-wide distributions. We have included this information in Table S3 and updated the figure legend of Fig 2 directing readers to the table.

The % mapped bisulfite reads seems a bit low (Table S7), considering the customized reference genome is used, and the high mRNA-seq mapping rates. Any possible reasons offered?

WGBS alignment rates are dependent on the repetitive nature of the genome under study and the length of reads (150 bp single-end for the replicate 1 and 150 bp paired-end in replicate 2). The rates we observed are consistent with what is observed in from methylome sequencing of the first reference genome for poplar (Nisqually) (Niederhuth, Bewick, et al, Genome Biology, 2016). We did observe a ~10% higher alignment rate to the Stettler reference compared to the rates when mapping this same WGBS data to the Nisqually reference and we did observe a higher rate for the paired-end reads, which fits expectations.

How the methylation levels are estimated should be clearly described, as well as how the bisulfite unconversion rate was estimated from lambda.

We have added a better description of how lambda DNA was used and how the methylation levels are defined.

Pg 12, if mCG and mCHG are cumulative across somatic development and point to a shared meristematic origin, what about mCHH? Is mCHH epimutation not accumulative at all? Also, on Fig 3C it seem there are few lines (regions) showing flat (non-accumulative), what are they and why?

Good question. CHH does not show significant accumulation for any genomic feature. To clarify this, we have added the following text to this section to make this clearer: "Cytosines in CHH context could not be shown to significantly accumulate epimutations (See table S9)."

To show the contrast between mCHH and mCHG/mCG, we have added the divergence of mCHH to Figure 3. The Table S8 (now S9) contains the rates and the respective p-values for evaluating whether the context/genomic feature shows an accumulation of epimutations different from the null model of no accumulation. For CHH, none of the genomic feature show a p-value below 0.99.

The almost flat line in Figure 3C shows the divergence in transposable elements, which accumulate very little in comparison to mCG and mCHG. We have amended the figures to include the dots of the divergence observations such that it is visible which line represents which genomic feature. This was included in our first submission, but during conversion to PDF the dots were somehow eliminated.

P12, L168, what are the estimates reported in *A. thaliana* MA lines?

We have modified the text as follows in response to this comment: "Remarkably, these estimates are very similar to those reported in *A. thaliana* MA lines, where the average CG and CHG rates are about 3.6×10^{-4} and 3.1×10^{-5} , respectively (Shahryar et al. 2019, co-submission)."

P13 L282, does the CMT expression vary with the gaining or losing methylation?

We examined the expression of CMT2 and CMT3. There is no variation among samples for CMT2. There is variation in expression of CMT3 and it is a differently expressed gene, however it is not correlated to methylation levels.

P13 L287, so the statement that "epimutations are not a result of biased reinforcement of DNA methylation during sexual reproduction or environment/genetic variation, but instead a feature of DNA methylation maintenance through mitotic cell divisions." sounds like a model rather than a solid conclusions? If so, the statement in Abstract should be tuned down. If it is more than a working model then perhaps more direct evidences should be expected/presented.

We agreed with the reviewer and as a result we have toned down our conclusions in the abstract and in the discussion.

P14 L298, I am not sure why the DMR are detected regardless the sequence contexts. Clearly CG, CHG and CHH have very different methylation levels ranging from 36% in mCG to 2% in

mCHH, and their behaviors in this study are apparently different. I would analyze the DMR by the sequence context rather than combining all.

Our goal in this type of analysis is to find regions targeted by multiple methylation pathways, as the genome of most angiosperms has either CG only methylation (gene body DNA methylation) or CG + non-CG methylation (either due to RdDM or CMT2). This type of analysis is consistent with our previous publications to differentiate these distinct methylated regions of the genome.

Not very clear on Fig 4a how to observe age-dependent accumulation of DMR

Please see response to point 5 by reviewer 1. In short, we failed to reference a supporting supplementary table of Figure 4a. This has now been remedied.

Are these DMR distributed randomly or there are hot spots? Since some of them prefer promoters, any specific genes are they targeting?

Based on GO enrichment analysis, DMRs within promoters are not enriched for any molecular functions or biological processes (weighted Fisher's exact, P value < 0.001). We have added this to the manuscript (line 324).

In DMR methods it say all samples are analyzed together to obtain DMR. If so, these are methylation variable regions among these samples. How does that compare to sets of DMR from pairwise comparisons between samples?

We do not perform pairwise analysis of samples for DMR detection, as methylpy was not designed for pairwise analyses. One of the biggest challenges with performing such analysis is that a significant fraction of regions will be lost after correcting for false positives accounting for multiple testing. In the case of our samples, the sample size for correction increases by 5,040 (7 factorial).

Also, are the direction of DMS (hyper or hypo) considered within one DMR? Do they have to be concordant? -- The Methods should cover all steps in details.

We apologize for not being clearer in the original submission. The program (methylpy) used in this study examines the direction (hyper vs hypo) in calculating DMSs. DMSs from different directions will not be linked. We have updated the methods section as follows:

Regions with at least three differentially methylated cytosines (DMS) were combined into raw DMRs. DMS with different directionality (hyper vs hypo) were not combined. Only DMRs that are at least 40-bp long with five or more cytosines (three of which are differentially methylated) with at least one read were used for subsequent analysis.

Perhaps more global analyses should be carried out for the allelic methylation changes in addition to reporting individual cases in Figure 4defg. How strong or prevalent are the allelic methylation changes?

This suggested analysis was performed and included in the original submission (Figure S6, now S7a). To improve the clarity of this analysis we have now added a heatmap as well (Figure S7b)

How do we see from Fig 5a that the accumulation of genetic and epigenetic changes are largely uncoupled? Both show accumulation, no?

We apologize for not being clearer. We have rephrased the interpretation of the results to the following: “These results show that divergence in DNA methylation is not accompanied by transcriptional divergence during tree aging, probably as a result of gene expression being much more dynamic and responsive to current environments.”

See response to reviewer 1 points 9 and 10.

Figure 5b, would it make more sense again to separate the CG, CHG, and CHG DMR here to study DMR-DEG correlation, as both the direction and the contexts matter?

In our opinion this doesn't make sense. It's well established in the field that in examples where there is differential methylation associated with differential gene expression that the methylation variation occurs in all cytosine contexts. This is the rationale for why we perform this type of analysis in this study and in our previously published studies.

Minor

The color key is missing on Fig 3abcd. I can't tell which line is which.

Thank you for pointing this out. It seems there was an error when our submission was converted to a PDF. Hopefully this is corrected now.

Second round of review

Reviewer 1

I have read the authors' responses and the revised manuscript. I think that their responses are quite reasonable and the revised manuscript is better than the original version. I have, nevertheless, questions below:

I think the most important finding in this paper is the neutral pattern of epimutation accumulation as the time goes on. To largely increase power of significant test for the neutral pattern, I have suggested (in my previous comments) to compare the epimutations accumulated between tree 13 and 14 (both are supposed to be from a single tree), because any new leaf from one tree is equally separated from the leaf of the other tree. Here only a simple logic is required: any new leaf is grown on a new shoot which should be from a new shoot in last year. Therefore, the equal divergence can be expected normally for the leave between two trees. Occasional, a new shoot could grow from a dormant bud. In this case, the equal divergence could not be expected. So this requirement is not depended on the specific branches of either tree. All samples from both trees can be used to test the neutrality hypothesis for a larger power. Here I could make an extreme example to show the simple logic. When there are two years old trees, which were derived from a single branch, any new leaf in tree 1 is expected to be equally separated from the leaf of tree 2, because there is no chance for a leaf developed from a dormant bud.

Though the authors mentioned that the studied tree was thought to be two independent trees, the very close proximity of two branches (as shown in the Fig. 1) indeed suggest they have a large chance to be from a single tree. However, since the poplar tree could generate asexually propagated clones from its root, even if the authors sampled another tree nearby, they still need to confirm whether they were independent meiotic products or asexual clones. Fortunately, the two closely related trees, no matter they are from a single tree or from asexual clones, still can be used to test the neutral pattern by the largely increased power. Actually, it is an easy way to increase the sample sizes for more clear test (Regarding the sample size $n=1$, it's certainly ideal to have more trees analyzed).

Reviewer 2

P.10/P11 The software/method PERTRAN by Shu is listed as unpublished. I thought the un-referenced data or sources are generally not acceptable for publication...

P. 19 Perhaps authors can discuss or explain the potential reasons for positive correlation between DMR-DEG (higher DNAm with higher gene expression), in addition to the 4 specific examples. The current conclusion sounds vague -- there is no global correlation (though the authors suspected the correlation between DNAm and transcription in Introduction), and yet the authors in P.19 did not want to rule out the correlation at specific loci. When observed unexpected positive correlation between DMR-DEG then no discussion or explanation is offered. These led audiences to nowhere with all the hard work the authors had done.

Minor:

less genic sequences

more tandem repeat sequences

Authors response

Reviewer #1: I have read the authors' responses and the revised manuscript. I think that their responses are quite reasonable and the revised manuscript is better than the original version. I have, nevertheless, questions below:

I think the most important finding in this paper is the neutral pattern of epimutation accumulation as the time goes on. To largely increase power of significant test for the neutral pattern, I have suggested (in my previous comments) to compare the epimutations accumulated between tree 13 and 14 (both are supposed to be from a single tree), because any new leaf from one tree is equally separated from the leaf of the other tree. Here only a simple logic is required: any new leaf is grown on a new shoot which should be from a new shoot in last year. Therefore, the equal divergence can be expected normally for the leave between two trees. Occasional, a new shoot could grow from a dormant bud. In this case, the equal divergence could not be expected. So this requirement is not depended on the specific branches of either tree. All samples from both trees can be used to test the neutrality hypothesis for a larger power. Here I could make an extreme example to show the simple logic. When there are two years old trees, which were derived from a single branch, any new leaf in tree 1 is expected to be equally separated from the leaf of tree 2, because there is no chance for a leaf developed from a dormant bud.

Though the authors mentioned that the studied tree was thought to be two independent trees, the very close proximity of two branches (as shown in the Fig. 1) indeed suggest they have a large chance to be from a single tree. However, since the poplar tree could generate asexually propagated clones from its root, even if the authors sampled another tree nearby, they still need to confirm whether they were independent meiotic products or asexual clones. Fortunately, the two closely related trees, no matter they are from a single tree or from asexual clones, still can be used to test the neutral pattern by the largely increased power. Actually, it is an easy way to increase the sample sizes for more clear test (Regarding the sample size $n=1$, it's certainly ideal to have more trees analyzed).

Response: The reviewer suggests to analyze both tree 13 and tree 14 together. They correctly point out that the clearest divergence should be expected when comparing epimutation divergence between tree13 and tree14, rather than within tree13 and tree14. Although we appreciate this thoughtful comment, they may have misread our analytical approach. We have already done what this reviewer suggests: We do analyze both trees together by treating them as a single tree with two main stems. For comparison we also analyzed both trees (tree13 and 14) separately. We observed a gain in power in the 'complete tree' analysis that this reviewer alludes to.

Reviewer #2: P.10/P11 The software/method PERTRAN by Shu is listed as unpublished. I thought the un-referenced data or sources are generally not acceptable for publication...

Response: The software/method by Shu is now published. We have modified the reference to this method as follows. Instead of stating it is unpublished, we have changed the text to read, "as described in Lovell et al, 2018).

P. 19 Perhaps authors can discuss or explain the potential reasons for positive correlation between DMR-DEG (higher DNAm with higher gene expression), in addition to the 4 specific examples. The current conclusion sounds vague -- there is no global correlation (though the authors suspected the correlation between DNAm and transcription in Introduction), and yet the authors in P.19 did not want to rule out the correlation at specific loci. When observed unexpected positive correlation

between DMR-DEG then no discussion or explanation is offered. These led audiences to nowhere with all the hard work the authors had done.

Response: As suggested by the reviewer we have modified the paragraph below copied from the main text to provide explanations about possible reasons for positive correlations between DMR-DEG.

“However, this global analysis does not rule out that DNA methylation changes at specific individual loci can have transcriptional consequences. To explore this in more detail, we analyzed DMRs proximal to DEGs, and correlated the methylation level of the DMR with the expression level of the gene. The correlation is positive when a higher methylation level in the DMR is associated with higher expression of the gene and negative when higher methylation is associated with lower expression of the gene. Regardless of where the DMR was located relative to the gene, we observed positive DMR-DEG correlations and negative DMR-DEG correlations. There was no bias for direction of correlation and genomic feature type (Fig. 5b). The negative correlations represent cases where DNA methylation is blocking cis-regulatory elements, whereas the positive correlations could represent cases where increased accessibility to transcriptional machinery of the gene leads to greater RdDM activity as observed previously (Secco et al, eLife).”